# CONVERGENCE ANALYSIS OF SPLIT LEARNING ON NON-IID DATA

## ABSTRACT

Split Learning (SL) is one promising variant of Federated Learning (FL), where the AI model is split and trained at the clients and the server collaboratively. By offloading the computation-intensive portions to the server, SL enables efficient model training on resource-constrained clients. Despite its booming applications, SL still lacks rigorous convergence analysis on non-IID data, which is critical for hyperparameter selection. In this paper, we first prove that SL exhibits an $\mathcal{O}(1/\sqrt{T})$ convergence rate for non-convex objectives on non-IID data, where $T$ is the number of total iterations. The derived convergence results can facilitate understanding the effect of some crucial factors in SL (e.g., data heterogeneity and local update steps). Comparing with the convergence result of FL, we show that the guarantee of SL is worse than FL in terms of training rounds on non-IID data. The experimental results verify our theory. Some generalized conclusions on the comparison between FL and SL in cross-device settings are also reported.

## 1 INTRODUCTION

Federating Learning (FL) is a popular distributed learning paradigm where multiple clients collaborate to train a global model under the orchestration of one central server. There are two settings in Federating Learning (FL) (McMahan et al., 2017) including (i) cross-silo where clients are organizations and the client number is typically less than 100 and (ii) cross-device where clients are Iot devices and the client number can be up to $10^{10}$ (Kairouz et al., 2021). To alleviate the computation bottleneck at resource-constrained IoT devices in the cross-device scenario, Split Learning (SL) (Gupta & Raskar, 2018; Vepakomma et al., 2018) splits the AI model to be trained at the clients and server separately. The computation-intensive portions are typically offloaded to the server, which is critical for the model training at resource-constrained devices. SL is regarded as one of the enabling technologies for edge intelligence in future networks (Zhou et al., 2019).

The comparisons of FL and SL are of practical interest for the design and deployment of intelligent networks. Existing studies focus on various aspects for their comparisons (Thapa et al., 2020; Gao et al., 2020; 2021), e.g., in terms of learning performance (Gupta & Raskar, 2018), computation efficiency (Vepakomma et al., 2018), communication overhead (Singh et al., 2019), and privacy issues (Thapa et al., 2021). For example, with the emphasis on the learning performance comparison, Gao et al. (2020; 2021) find that SL exhibits (i) faster convergence speed than FL under IID data in terms of communication rounds; (ii) better learning performance under imbalanced data; (iii) worse learning performance under (extreme) non-IID data, etc. The difference arises from the distinct process of model updates of FL and SL. In particular, FL takes the average of the local model parameters at the end of each round; SL only trains the clients in sequence and does not average the client updates. Figure 1 plots the client drift (Karimireddy et al., 2020; Wang et al., 2020; Li et al., 2022) of FL and SL under both IID and non-IID data to visualize the update process. Under the IID setting, SL approach the global optima $\mathbf{x}^*$ faster than FL given the sequential training mechanism. In contrast, under the non-IID setting, SL may deviate from the global optima for the same reason.

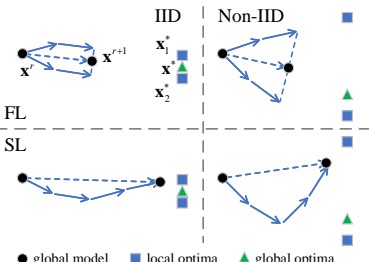

Figure 1: Illustration of the model updates of FL and SL for 2 clients and 2 local update steps during one round.

Convergence analysis is critical for the performance comparison between SL and FL. Specifically, a rigorous analysis is of paramount importance for the vital research questions raised by (Gao et al., 2020) (which are only empirically evaluated but remain unsolved in theory): RQ1-"*What factors affect SL performance?*" and RQ2-"*In which setting will the SL performance outperform FL?*". A wealth of work has analyzed the convergence of FL in the cases of IID (Stich, 2018; Zhou & Cong, 2017; Khaled et al., 2020), non-IID (Li et al., 2020; 2019; Khaled et al., 2020; Karimireddy et al., 2020) and unbalanced data (Wang et al., 2020). However, with the distinct update process, the convergence analysis of SL has yet to be solved on non-IID data. To this end, this paper first derives a rigorous convergence results of SL and draw the comparison results of FL and SL theoretically.

**Main contributions.** The main contributions can be summarized with respect to the two research questions above:

- We prove the convergence of SL on non-IID data with the standard assumptions used in FL literature [1] with a convergence rate of $\mathcal{O}(1/\sqrt{T})$ in Section 4.2. By this, we find that the convergence of SL is affected by factors such as data heterogeneity and the number of local update steps. Experimental results verify the analysis results empirically in Section 5.2. To the best of our knowledge, this work is the first to give the convergence analysis of SL on non-IID data.

- We compare FL[2] and SL in theory (Section 4.3) and in practice (Section 5.3). Theoretically, the guarantee of SL is worse than FL in terms of training rounds on non-IID data. Empirically, we provide some generalized conclusions of FL and SL in cross-device settings, including **(i)** the best and threshold learning rate of SL is smaller than FL; **(ii)** the performance of SL is worse than FL when the number of local update steps is large on highly non-IID data; **(iii)** the performance of SL can be better than FL when choosing small the number of local update steps is small on highly non-IID data.

## 2    PRELIMINARIES AND ALGORITHM OF SPLIT LEARNING

As two of the most popular distributed learning frameworks, both FL and SL aim to train the global model from distributed datasets. The optimization problem of FL and SL with $N$ clients can be given by

$$\min_{\mathbf{x}\in\mathbb{R}^d}\left\{f(\mathbf{x}) := \sum_{i=1}^{N} p_i f_i(\mathbf{x})\right\}, \tag{1}$$

where $p_i = n_i/n$ is the ratio of local samples at client $i$ ($n$ and $n_i$ are the sizes of the overall dataset $\mathcal{D}$ and local dataset $\mathcal{D}_i$ at client $i$, respectively), $\mathbf{x}$ is the model parameters, $f(\mathbf{x})$ is the global objective, $f_i(\mathbf{x})$ denotes the local objective function on client $i$. In particular, $f_i(\mathbf{x}) := \mathbb{E}_{\xi_i \sim \mathcal{D}_i}[f_i(\mathbf{x}; \xi_i)] = \frac{1}{n_i}\sum_{\xi_i \in \mathcal{D}_i} f_i(\mathbf{x}; \xi_i)$, where $\xi_i$ represents a data sample from the local dataset $\mathcal{D}_i$.

**SL with the global learning rate.** The relay-based (sequential) training process across clients makes SL significantly different (from FL), which has been described in Algorithm 1. Considering the massive number of clients in cross-device setting, only a subset $\mathcal{S}$ of clients are selected for model training at each round. The update order of the selected clients can be meticulously designed or randomly determined (used in this paper). The $i$-th client requests and initializes with the lasted model (step 4) and then performs multiple local updates (step 5-11)[3]. After $K$ local updates, the client will send the model parameters to the next client (i.e., the $i + 1$-th client). The local update

---

[1]We only show the convergence for non-convex objective functions here since SL is now often used in large deep learning models whose objective functions are possibly non-convex. Nevertheless, similar methods can be used to get the convergence for general convex functions and strongly convex functions.

[2]`FedAvg` is used for comparison in this work.

[3]The client and the server cooperate to conduct the local updates. Note that in SL, though the model update requires communication between the client and server, **the model is still trained on the local dataset**. So the process is called *local update* (Thapa et al., 2020). The concatenation of the client-side and server-side models after each local update is called *local model*.

process can be stated as:

$$\text{Initialize:} \quad \mathbf{x}_i^{(r,0)} = \begin{cases} \mathbf{x}^r, i = 1 \\ \mathbf{x}_{i-1}^{(r,K)}, i > 1 \end{cases} \tag{2}$$

$$\text{Update:} \quad \mathbf{x}_i^{(r,k+1)} = \mathbf{x}_i^{(r,k)} - \eta_l \mathbf{g}_i(\mathbf{x}_i^{(r,k)}), \tag{3}$$

where $\mathbf{x}_i^{(r,k)}$ denotes the complete local model (parameters) after the $k$-th local update on the local dataset $\mathcal{D}_i$ in the $r$-th round, $\eta_l$ is the local learning rate and $\mathbf{g}_i(\mathbf{x}_i^{(r,k)}) := \nabla f_i(\mathbf{x}_i^{(r,k)}; \xi_i^{(r,k)})$ represents the stochastic gradients over on the mini-batch $\xi_i^{(r,k)}$ sampled randomly from $\mathcal{D}_i$. Note that the complete model $\mathbf{x}_i^{(r,k)}$ consists of the client-side model $\mathbf{x}_{c,i}^{(r,k)}$ and server-side model $\mathbf{x}_{s,i}^{(r,k)}$, as shown in Algorithm 1. The clients and server update their models synchronously and without loss of generality, we do not highlight the model locations in the following.

After the last client in $\mathcal{S}$ (i.e., the $|\mathcal{S}|$-th client) completes its local updates, it will request the initial client-side model of the current round. The global update is conducted in both the client and server:

$$\mathbf{x}^{r+1} = \mathbf{x}^r + \eta_g(\mathbf{x}_{|\mathcal{S}|}^{(r,K)} - \mathbf{x}^r), \tag{4}$$

where $\mathbf{x}^r$ denotes the complete global model (parameters) in the $r$-th round ($\mathbf{x}^r$ equals $\mathbf{x}_i^{(r,0)}$ only when $i = 1$, which is different from FL), $\eta_g$ is the global learning rate. Note in Eq. (4) that we propose adding the global learning rate $\eta_g$ against the vanilla SL algorithm (Gupta & Raskar, 2018; Vepakomma et al., 2018; Thapa et al., 2020; 2021). The global learning rate design mechanism is originally developed in the FL setting (Karimireddy et al., 2020; Reddi et al., 2020; Wang et al., 2020) to reduce the client drift, and can be readily adopted in SL for the same function. Algorithm 1 can operate in both the centralized and peer-to-peer mode (see Appendix A) and is reduced to the vanilla SL if steps 13-14 removed. For brevity, we have omitted some unconcerned details of SL in Algorithm 1, e.g., the security and privacy settings. More details about SL can be found in Appendix A or Gupta & Raskar (2018); Thapa et al. (2021).

---

**Algorithm 1** Split Learning with the Global Learning Rate

---

    **Some notations**:
    – $\mathbf{x}_c$ denotes the client-side model (parameters)
    – $\mathbf{x}_s$ denotes the server-side model (parameters)
    – $\mathbf{x} := [\mathbf{x}_c, \mathbf{x}_s]$ denotes the complete model (parameters)
1: **for** round $r = 0, \ldots, R - 1$ **do**
2:     Sample a subset $\mathcal{S}$ of clients and determine their update order
3:     **for** client index $i = 1, \ldots, |\mathcal{S}|$ **do**
4:         **Client $i$:** Request the latest client-side model and initialize $\mathbf{x}_{c,i}^{(r,0)} \leftarrow \begin{cases} \mathbf{x}_c^r, i = 1 \\ \mathbf{x}_{c,i-1}^{(r,K)}, i > 1 \end{cases}$
5:         **for** local update step $k = 0, \ldots, K - 1$ **do**
6:             **Client $i$:** Forward propagation and send activations to the server
7:             **Server:** Forward propagation with activations from the client
8:             **Server:** Back-propagation and send gradients to the client
9:             **Client $i$:** Back-propagation with gradients from the server
10:             Local model updates: $\begin{cases} \text{Client } i: \quad \mathbf{x}_{c,i}^{(r,k+1)} \leftarrow \mathbf{x}_{c,i}^{(r,k)} - \eta_l \mathbf{g}_i(\mathbf{x}_{c,i}^{(r,k)}) \\ \text{Server:} \quad \mathbf{x}_{s,i}^{(r,k+1)} \leftarrow \mathbf{x}_{s,i}^{(r,k)} - \eta_l \mathbf{g}_i(\mathbf{x}_{s,i}^{(r,k)}) \end{cases}$
11:         **end for**
12:     **end for**
13:     **Client $|\mathcal{S}|$:** Request the initial client-side model $\mathbf{x}_c^r$ of the current round
14:     Global model updates: $\begin{cases} \text{Client } |\mathcal{S}|: \quad \mathbf{x}_c^{r+1} \leftarrow \mathbf{x}_c^r + \eta_g(\mathbf{x}_{c,|\mathcal{S}|}^{(r,K)} - \mathbf{x}_c^r) \\ \text{Server:} \quad \mathbf{x}_s^{r+1} \leftarrow \mathbf{x}_s^r + \eta_g(\mathbf{x}_{s,|\mathcal{S}|}^{(r,K)} - \mathbf{x}_s^r) \end{cases}$
15:     **Client $|\mathcal{S}|$ and Server:** Store the global model
16: **end for**

---

## 3 RELATED WORK

**Variants of SL.** SL is deemed as a promising paradigm for distributed model training at resource-constrained devices, given its computational efficiency on the client side. Most existing works focus on reducing the training delay arising from the relay-based training manner in the multi-user scenario. `SplitFed` (Thapa et al., 2020) is one popular model parallel algorithm that combines the strengths of FL and SL, where each client has one corresponding instance of server-side model in the *main server* to form a pair. Each pair constitutes a complete model and conducts the local update in parallel. After each training round, the *fed server* collects and aggregates on the clien-side local updates. The aggregated client-side model will be disseminated to all the clients before next round. The *main server* does the same operations to the instances of the server-side model. `SplitFedv2` (Thapa et al., 2020), `SplitFedv3` (Gawali et al., 2021) and `SFLG` (Gao et al., 2021), `FedSeq` (Zaccone et al., 2022) are the variants of `SplitFed`. In particular, `SFLG` (Gao et al., 2021) is one generalized variants of `SplitFed`, combining `SplitFed` and `SplitFedv2`.

**Convergence analysis of SL.** In the case of IID data, the convergence analysis of SL is identical to standard Minibatch-SGD (Wang et al., 2022; Park et al., 2021), so some convergence properties of Minibatch SGD is applied to SL too. The algorithm in Han et al. (2021) reduces the latency and downlink communication on `SplitFed` by adding auxiliary networks at client-side for quick model updates. Their convergence analysis combines the analysis of Belilovsky et al. (2020) and `FedAvg`. Wang et al. (2022) proposed `FedLite` to reduce the uplink communication overhead by compressing activations with product quantization and provided the convergence analysis of `FedLite`. However, their convergence recovers that of Minibatch SGD when there is no quantization. SGD with biased gradients (Ajalloeian & Stich, 2020) is also related. However, it only converges to a neighborhood of the solution. Furthermore, we find that Woodworth et al. (2020a;b) compared the convergence of distributed Minibatch SGD and local SGD under homogeneous and heterogeneous settings, respectively. To differentiate our work, we show how these algorithms operate in Appendix B. As a result, the convergence of SL on non-IID data is still lacking.

## 4 CONVERGENCE ANALYSIS OF SL

### 4.1 ASSUMPTIONS

We make the following standard assumptions on the local objective functions $\{f_i(\mathbf{x})\}_{i=1}^N$.

**Assumption 1** (*L*-smooth). *Each local objective function $f_i$ is L-smooth, $i \in \{1, 2, \ldots, N\}$, i.e., there exists a constant $L > 0$ such that $\|\nabla f_i(\mathbf{x}) - \nabla f_i(\mathbf{y})\| \le L \|\mathbf{x} - \mathbf{y}\|$ for all $\mathbf{x}$ and $\mathbf{y}$.*

**Assumption 2** (Unbiased gradient and bounded variance). *For each client $i$, $i \in \{1, 2, \ldots, N\}$, the stochastic gradient $\mathbf{g}_i(\mathbf{x}) := \nabla f_i(\mathbf{x}; \xi_i)$ is unbiased $\mathbb{E}[\mathbf{g}_i(\mathbf{x})] = \nabla f_i(\mathbf{x})$ and has bounded variance $\mathbb{E}_{\xi_i}[\|\mathbf{g}_i(\mathbf{x}) - \nabla f_i(\mathbf{x})\|^2] \le \sigma^2$.*

**Assumption 3** (Bounded dissimilarity). *There exist constants $B \ge 1$ and $G \ge 0$ such that $\frac{1}{N} \sum_{i=1}^N \|\nabla f_i(\mathbf{x})\|^2 \le B^2 \|\nabla f(\mathbf{x})\|^2 + G^2$. In the IID case, $B = 1$ and $G = 0$, since all the local objective functions are identical to each other. In the non-IID case, $B$ and $G$ measure the heterogeneity of data distribution.*

### 4.2 CONVERGENCE RESULT AND DISCUSSION

Without loss of generality, all the clients participate in the SL process ($|\mathcal{S}| = N$) and the unweighted global objective function $f(\mathbf{x}) = \frac{1}{N} \sum_{i=1}^N f_i(\mathbf{x})$ is adopted. Note that the results in the unweighted case can be readily extended to the weighted case of Eq. (1). Following the proof of Wang et al. (2020); Karimireddy et al. (2020); Khaled et al. (2020), we give the convergence results of SL (details are in Appendix D), as follows:

**Theorem 1.** *Let Assumptions 1, 2 and 3 hold. Suppose that the local learning rate satisfies $\eta_l \leq \frac{1}{2NKL} \min \left\{ \frac{1}{\sqrt{2B^2+1}}, \frac{1}{\eta_g} \right\}$. For Algorithm 1, it holds that*

$$\mathbb{E}[\|\nabla f(\bar{\mathbf{x}}^R)\|^2] \leq \underbrace{\frac{4[f(\mathbf{x}^0) - f(\mathbf{x}^*)]}{NK\eta_g\eta_l R}}_{T_1:\text{initialization error}} + \underbrace{12N^2K^2\eta_l^2L^2G^2 + 6N^2K\eta_l^2L^2\sigma^2}_{T_2:\text{client drift error}} + \underbrace{4N\eta_g\eta_lL\sigma^2}_{T_3:\text{global variance}}, \quad (5)$$

*where $\bar{\mathbf{x}}^R = \frac{1}{R}\sum_{r=0}^{R-1}\mathbf{x}^r$ is the averaged global model over the $R$ rounds.*

**Corollary 1.** *Choose $\eta_g\eta_l = \frac{1}{L\sqrt{T}}$ and apply the result of Theorem 1. For sufficiently large $T$ ($T \geq 4N^2K^2 \max\left\{\frac{2B^2+1}{\eta_g^2}, 1\right\}$), it holds that*

$$\mathbb{E}[\|\nabla f(\bar{\mathbf{x}}^R)\|^2] \leq \underbrace{\mathcal{O}\left(\frac{L[f(\mathbf{x}^0)-f(\mathbf{x}^*)]}{\sqrt{T}}\right)}_{T_1:\text{initialization error}} + \underbrace{\mathcal{O}\left(\frac{N^2K^2G^2 + N^2K\sigma^2}{\eta_g^2T}\right)}_{T_2:\text{client drift error}} + \underbrace{\mathcal{O}\left(\frac{\sigma^2}{\sqrt{T}}\right)}_{T_3:\text{global variance}}, \quad (6)$$

*where $T$ is the total number of iterations (i.e., $NKR$ in SL).*

The upper bound of $\mathbb{E}[\|\nabla f(\bar{\mathbf{x}}^R)\|^2]$ consists of three types of terms: (i) initialization error, (ii) client drift error, caused by the client drift (see Lemma 4 in Appendix D.2), (iii) global variance. We can see that the result demonstrates the relationships between convergence and factors such as the number of clients, the data heterogeneity, the global/local learning rates and the local update steps. According to the result, a large $\eta_l$ means higher rate that the initialization error decreases at but causes large client drift error and global variance. Next, we discuss the convergence rate and the influence factors of SL in detail.

**Convergence rate.** By Corollary 1, for sufficiently large $T$, the convergence rate is determined by the initialization error and global variance (see Eq. (6)), resulting in a convergence rate of $\mathcal{O}(1/\sqrt{T})$. We can recover the convergence of SGD (Bottou et al., 2018; Wang et al., 2020) when $N = 1$ and $K = 1$ (i.e., without the client drift error) — It is true but not shown in Theorem 1 directly, since it is complicated to write the constant details in Eq. (5). We defer the discussion to Appendix D.5.

**Effect of data heterogeneity.** According to Theorem 1, when on highly non-IID data ($B$ and $G$ are large), a small $\eta_l$ is required for the convergence of SL (see the condition of Theorem 1). In addition, the client drift error also increases. As a result, large data heterogeneity harms the convergence of SL, which is consistent with the previous study (Gao et al., 2020; 2021).

**Effect of $K$.** $K$ is the number of local update steps. An immediate question is whether we can improve the convergence by adding local update steps when $R$ is fixed. The answer is yes. By Eq. (5), as $K$ increases, the initialization error decreases and the client drift error increases, which implies that the optimal $K$ exists. We can further get that larger data heterogeneity makes the optimal $K$ smaller based on Eq. (5). This property is analogous to FL (McMahan et al., 2017).

**Effect of $\eta_g$.** The global learning rate $\eta_g$ can be used to reduce the client drift without hurting the progress of SL. A large $\eta_g$ reduces the client drift error, hence improving the convergence rate. For example, by Corollary 1, SL shows $\mathcal{O}(\frac{LF+\sigma^2}{\sqrt{T}} + \frac{N^2K^2G^2+N^2K\sigma^2}{T})$ convergence rate if $\eta_g = 1$ ($F := f(\mathbf{x}^0) - f(\mathbf{x}^*)$); while it shows a convergence rate of $\mathcal{O}(\frac{LF+\sigma^2}{\sqrt{T}} + \frac{K^2G^2+K\sigma^2}{T})$ if $\eta_g = N$.

Some detailed analysis of these factors are deferred to Section 5.2, combined with the experiments.

### 4.3 COMPARISON BETWEEN FL AND SL

In this section, we first give the convergence result of FL based on our setting, and then compare the results of FL and SL to answer the second question "*In which setting will the SL performance outperform FL?*" theoretically.

We summarize the convergence results of FL ((Wang et al., 2020), reproduced by Theorem 2 in Appendix D.3) and SL (Theorem 1) in Table 1. The convergence guarantee of FL is one of the best known convergence results for the non-convex functions of FL, which makes our comparison

Table 1: Comparison of convergence results between FL (Wang et al., 2020) and SL (Theorem 1) for non-convex functions. The convergence guarantee of FL is given in the 1-st (upper bound) and 2-th (constraints on $\eta_l$) rows. The effective learning rate versions are shown in the 3-rd (FL) and 5-th (SL) row. The number of rounds required to reach $\epsilon$ accuracy is given in the 4-th (FL) and 6-th (SL) rows. $F := f(\mathbf{x}^0) - f(\mathbf{x}^*)$. Constants (including $L$) are omitted. $\eta_g$ is set to $\eta_g = 1$.

| | |
|---|---|
| **FL** (Wang et al., 2020) | $\mathcal{O}\left(\frac{F}{\eta_l K R}\right) + \mathcal{O}\left(\eta_l^2 K^2 G^2 + \eta_l^2 K \sigma^2\right) + \mathcal{O}\left(\frac{\eta_l \sigma^2}{N}\right)$ |
| | Constraint: $\eta_l \leq \frac{1}{2KL}\min\left\{\frac{1}{\sqrt{2B^2+1}}, \frac{1}{\eta_g}\right\}$ |
| | $\tilde{\eta}_{\text{FL}}$: $\mathcal{O}\left(\frac{F}{\tilde{\eta}_{\text{FL}} R}\right) + \mathcal{O}\left(\tilde{\eta}_{\text{FL}}^2 G^2 + \frac{\tilde{\eta}_{\text{FL}}^2 \sigma^2}{K}\right) + \mathcal{O}\left(\frac{\tilde{\eta}_{\text{FL}} \sigma^2}{NK}\right)$ |
| | $R_\epsilon = \mathcal{O}\left(\frac{F^2}{\epsilon^2} + \frac{\sigma^4}{N^2 K^2 \epsilon^2} + \frac{KG^2 + \sigma^2}{K\epsilon}\right)$ |
| **SL** | $\tilde{\eta}_{\text{SL}}$: $\mathcal{O}\left(\frac{F}{\tilde{\eta}_{\text{SL}} R}\right) + \mathcal{O}\left(\tilde{\eta}_{\text{SL}}^2 G^2 + \frac{\tilde{\eta}_{\text{SL}}^2 \sigma^2}{K}\right) + \mathcal{O}\left(\frac{\tilde{\eta}_{\text{SL}} \sigma^2}{K}\right)$ |
| | $R_\epsilon = \mathcal{O}\left(\frac{F^2}{\epsilon^2} + \frac{\sigma^4}{K^2 \epsilon^2} + \frac{KG^2 + \sigma^2}{K\epsilon}\right)$ |

persuasive. Choosing $\eta_l = \frac{\sqrt{N}}{L\sqrt{T}}$ for Theorem 2 provides the $\mathcal{O}(1/\sqrt{NT})$ convergence rate, the linear speedup in FL.

**The constraint on the local learning rate of SL is tougher than FL.** The local learning rate of SL (see Theorem 1) has tougher constraints than FL (see the 2-nd row of Table 1), which indicates SL is more sensitive to the heterogeneity of data. This is significant for the selection of the learning rate of SL in practice (see the comparison experiments).

**Effective learning rate.** We next focus on comparing FL and SL in terms of rounds. Note that this comparison (running for the same $R$) is fair given the same total computation cost (including the computation cost on client-side and server-side). Beginning with the observation that the convergence guarantees and constraints seem very alike if choosing $\eta_{l(\text{SL})} = \eta_{l(\text{FL})}/N$ — $\eta_{l(\text{FL})}$ and $\eta_{l(\text{SL})}$ denote the local learning rate of FL and SL respectively, we define the effective learning rate $\tilde{\eta}_{\text{FL}} := K\eta_g\eta_l$ for FL and $\tilde{\eta}_{\text{SL}} := NK\eta_g\eta_l$ for SL as Karimireddy et al. (2020); Wang et al. (2020) did. Note that the effective learning rate is unequal for FL and SL. As a result, we obtain the convergence guarantee of the effective learning rate version exhibited in the 3-rd and 5-th rows in Table 1.

**The guarantee of SL is worse than FL in terms of training rounds on non-IID data.** To make a comparison, we need to choose appropriate $\eta_l$ for both. Considering $\tilde{\eta}_{\text{FL}}$ and $\tilde{\eta}_{\text{SL}}$ has the same constraints, we can choose $\tilde{\eta}_{\text{FL}} = \tilde{\eta}_{\text{SL}} = 1/\sqrt{R}$ for both bounds (see the 3-rd and 5-th rows in Table 1). Then we get: (i) $\mathcal{O}\left(\frac{F}{\sqrt{R}} + \frac{KG^2+\sigma^2}{KR} + \frac{\sigma^2}{NK\sqrt{R}}\right)$ of FL and (ii) $\mathcal{O}\left(\frac{F}{\sqrt{R}} + \frac{KG^2+\sigma^2}{KR} + \frac{\sigma^2}{K\sqrt{R}}\right)$ of SL after $R$ rounds, i.e., the round complexity $R_\epsilon$ shown in 4-th and 6-th rows in Table 1. Then for sufficiently large $R$, $R_\epsilon$ is determined by the first and the second term (see the 4-th and 6-th rows). In particular, the only difference in the complexity appears in the second term (we have marked it with red), which indicates that the guarantee of SL is worse than FL in terms of rounds. However, we note that the gap is not obvious when $K$ or $\sigma$ is small. Further, considering the constants advantage of SL (It is true but not shown in Theorem 1, see Appendix D.2), **the performance comparison between FL and SL is still uncertain**. To make a more detailed comparison, we have conducted adequate experiments and given some generalized conclusions in Section 5.3.

## 5 EXPERIMENTS

Our experiment environment is ideal (without the communication and computation restrictions), nevertheless, is enough to examine our convergence theory. The convergence rate is evaluated in terms of the training rounds. We demonstrate the detailed experimental setup in Section 5.1, evaluate

the effect of the factors on the performance of SL in Section 5.2, and compare FL and SL in cross-device settings in Section 5.3. More experimental details are in Appendix E.

## 5.1 EXPERIMENTAL SETUP

**Datasets and models.** We adopt the following setups: (i) training LeNet-5 (LeCun et al., 1998) on the MNIST dataset (LeCun et al., 1998); (ii) training LeNet-5 on the Fashion-MNIST dataset (Xiao et al., 2017); (iii) training VGG-11 (Simonyan & Zisserman, 2014) on the CIFAR-10 dataset (Krizhevsky et al., 2009). For SL, the LeNet-5 is split after the second 2D MaxPool layer, with 6% of the entire model size retained in the client; the VGG-11 is split after the third 2D MaxPool layer, with 10% of the entire model size at the client. Ideally, the split layer position has no effect on the performance of SL (Wang et al., 2022).

**Data distribution.** Both IID and non-IID datasets are considered in the experiments. For the non-IID setting, we adopt two mechanisms to generate non-IID data: (i) using a Dirichlet distribution $\text{Dir}(\alpha)$ to generate mildly non-IID data, where a smaller $\alpha$ indicates higher data heterogeneity (Hsu et al., 2019; Zhu et al., 2021); (ii) using a similar mechanism like (McMahan et al., 2017; Zhao et al., 2018) to generate the pathological non-IID data, e.g., one distribution in which most clients only contain samples from 2 (5) classes, denoted as $C = 2\,(5)$. Note that the data distribution generated by the first mechanism is unbalanced and the second is balanced. We will use "$\alpha = \#$" and "$C = \#$" to represent the different data distributions in the following, where $\#$ is the parameter.

## 5.2 EXPERIMENTAL RESULTS OF SL

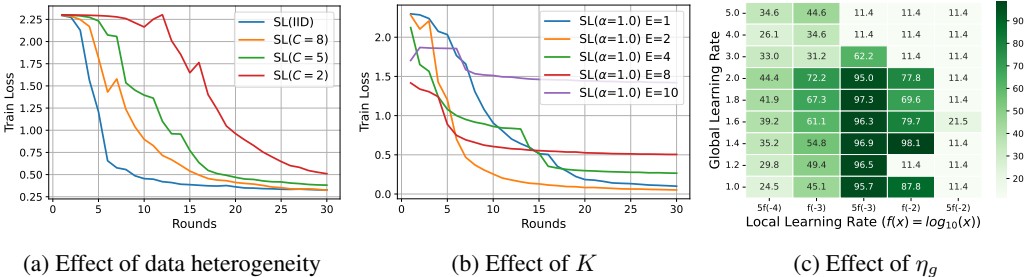

(a) Effect of data heterogeneity          (b) Effect of $K$          (c) Effect of $\eta_g$

Figure 2: Results of SL on MNIST dataset.

In this section, we study the effects of data heterogeneity local update steps using the MNIST dataset. The training samples are assigned to 10 clients. For data heterogeneity, we use four distributions: IID, $C = 8, 5, 2$. For $K$, we use five settings, $E = 1, 2, 4, 8, 10$ over $\text{Dir}_{10}(1.0)$ distribution. $E$ is the local epochs (McMahan et al., 2017), which satisfies $K = \max\{En_i/b\}$, where $b$ is the mini-batch size. So $E$ can measure the value of $K$ with $n_i$ and $b$ fixed. The results on Fashion-MNIST and CIFAR-10 are deferred to Appendix E.1.

**Effect of data heterogeneity.** As shown in Figure 2a, the training loss curve of IID distribution is the lowest and most stable. When the data heterogeneity increases ($C$ decreases from 8 to 2), SL shows worse performance. The phenomenon is in accordance with our analysis that large data heterogeneity harms the convergence of SL in Section 4.2.

**Effect of $K$.** Figure 2b shows the training loss in terms of communication rounds $R$. SL shows the best performance when $E = 2$. This verifies that optimal $K$ exists and suitable $K$ can improve the convergence. Over-large $K$ can even harm the convergence rate (see curves $E = 8$ and $E = 10$).

**Effect of $\eta_g$.** The effect of $\eta_g$ in FL has been empirically studied in Reddi et al. (2020). They tune $\eta_l$ and $\eta_g$ by grid search. We follow a similar method to study $\eta_g$ of SL by choosing different combinations of $\eta_g$ and $\eta_l$. As shown in Figure 2c, the dark green grids with high test accuracy concentrate on the left bottom triangular regions, which shows that $\eta_g \eta_l$ should avoid being too large. Also note that $\eta_g$ cannot be set infinitely too large and has a limited range, which is also observed in FL (Reddi et al., 2020). Thus tuning $\eta_g$ in the limited range is suggested. Besides, we find that $\eta_g$ can be introduced to other SL frameworks, like `SplitFed`. However, further research on $\eta_g$ is needed to address the issues such as *how to tune $\eta_g$ to gain improvement?* in FL and SL.

### 5.3 EMPIRICAL COMPARISON BETWEEN FL AND SL

We have compared FL and SL theoretically in Section 4.2. The question arises that *how about the learning performance of SL in practice compared to FL?* In the previous work, the same learning rates are used to evaluate the performances of SL and FL (Gao et al., 2020; 2021). However, theoretical analyses in Section 4.2 show that the appropriate learning rate for SL may deviate from that of FL. Thus, we evaluate the performance of SL and FL with different learning rates for fair comparison. The learning rates are selected from {0.0005, 0.001, 0.005, 0.01, 0.05, 0.1}. To evaluate the effect of $K$ on the performance comparison of FL and SL, we adopt two settings: $E = 1$ and $E = 10$. We run 1000 rounds of training on MNIST, Fashion-MNIST and 4000 rounds on CIFAR-10 dataset when $E = 1$; run 100 rounds of training on MNIST, Fashion-MNIST and 400 rounds on CIFAR-10 dataset when $E = 10$. The average top-1 test accuracy over the last 10% rounds are shown in Table 2, e.g., the average accuracy over the last 400 rounds of the total 4000 rounds on CIFAR-10 when $E = 1$.

**Cross-device setting.** We compare the performance of FL and SL in cross-device settings, where the client number is enormous and the local dataset size is small. Specifically, (i) MNIST: the training data is split into 1000 clients; (ii) Fashion-MNIST: the training data is split into 1000 clients; (iii) CIFAR-10: the training data is split into 500 clients. The number of samples per client depends on the data distribution, e.g., 60 (MNIST), 60 (Fashion-MNIST) and 100 (CIFAR-10) samples per client in the second partition mechanism. The mini-batch size is 10 for all setups under the consideration of low computation power of IoT devices. The original test sets are used to evaluate the generalization performance of the global model (test accuracy) after each training round.

**The best and threshold learning rate of SL is smaller than FL.** We refer to the learning rate making the best test accuracy and minimal learning rate making the training die as the best learning rate and the threshold learning rate. According to our theory, the tougher constraint indicates the smaller threshold learning rate of SL and the math property of Eq. (5) indicates a smaller best learning rate. To verify this point, we use the "best" learning rate, which makes the "best" test accuracy among the learning rates we choose, to substitute the actual best learning rate. The "threshold" learning rate is defined alike. As shown in Table 2, the "best" and "threshold" learning rates of SL are smaller than FL, especially on highly non-IID data (e.g., $\alpha = 0.2$). Furthermore, we note that the "best" and "threshold" learning rate turn small as the heterogeneity of data becomes large, which is also in accordance with our theory.

**Performance comparison on IID data.** SL can obtain a faster convergence rate with comparable or even higher test accuracy than FL (see the left plot in Figure 3 and Table 2). This is identical to the conclusion in Gao et al. (2020; 2021).

**Performance comparison on non-IID data.** Our theory proves that the guarantee of SL is worse than FL in terms of rounds when $K$ is large. As shown in Table 2, almost all the "best" test accuracy of FL on very highly non-IID data (i.e., $\alpha = 0.2$ and $C = 2$) beats that of SL when $E = 10$. When $E$ ($K$) is large, FL converges faster than SL too (see the middle plot in Figure 3). However, we find that SL has a better performance than FL when $E$ ($K$) is small (see the $E = 1$ column in Table 2 and the right plot in Figure 3). Even in some cases ($\alpha = 0.5$ and $C = 5$ on CIFAR-10) when $E = 10$, SL is better.

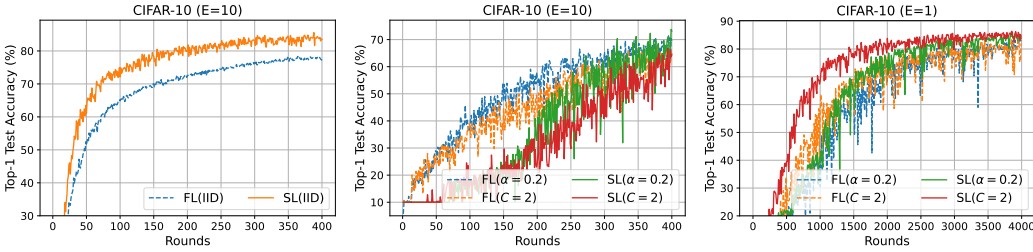

Figure 3: Top-1 test accuracy of FL and SL on non-IID data in terms of rounds. We illustrate some results of CIFAR-10 from Table 2.

Table 2: Performance comparison between FL and SL in cross-device settings. The "best" test accuracy (%) is in the *"Best" accuracy (lr)* column (the "best" learning rate is in the parenthesis) and the "threshold" learning rate in the *"Threshold" lr* column. Note that ># means that the "threshold" learning rate is larger than #. The higher "best" accuracy between FL and SL is marked in bold (excluding the results whose difference is within 1%).

| | Dataset | Distribution | $E=1$ | | | | $E=10$ | | | |
| | | | "Best" accuracy (lr) | | "Threshold" lr | | "Best" accuracy (lr) | | "Threshold" lr | |
| | | | FL | SL | FL | SL | FL | SL | FL | SL |
|---|---|---|---|---|---|---|---|---|---|---|
| 1 | | IID | 99.0 (0.05) | 98.9 (0.005) | > 0.1 | 0.05 | 97.9 (0.01) | 97.6 (0.005) | 0.1 | 0.05 |
| 2 | | $\alpha = 5.0$ | 99.1 (0.05) | 98.9 (0.01) | > 0.1 | 0.05 | 98.0 (0.01) | 97.8 (0.001) | 0.05 | 0.01 |
| 3 | | $\alpha = 0.5$ | 98.9 (0.05) | 98.8 (0.005) | > 0.1 | 0.05 | 97.9 (0.01) | 97.9 (0.001) | 0.05 | 0.005 |
| 4 | MNIST | $\alpha = 0.2$ | 98.7 (0.05) | 98.8 (0.005) | 0.1 | 0.05 | 96.8 (0.01) | 96.8 (0.001) | 0.05 | 0.01 |
| 5 | | $C = 5$ | 99.0 (0.05) | 98.9 (0.005) | > 0.1 | 0.05 | 98.0 (0.01) | 98.0 (0.001) | 0.05 | 0.01 |
| 6 | | $C = 2$ | 98.8 (0.05) | 99.0 (0.01) | > 0.1 | 0.05 | 96.8 (0.005) | 97.1 (0.001) | 0.05 | 0.005 |
| 7 | | IID | 88.1 (0.05) | 88.4 (0.005) | > 0.1 | 0.1 | **85.0 (0.01)** | 84.0 (0.001) | 0.1 | 0.05 |
| 8 | | $\alpha = 5.0$ | 88.2 (0.05) | 88.8 (0.005) | > 0.1 | 0.1 | **84.5 (0.01)** | 83.2 (0.0005) | 0.05 | 0.05 |
| 9 | Fashion- | $\alpha = 0.5$ | 86.7 (0.05) | 87.6 (0.005) | > 0.1 | 0.1 | **83.8 (0.01)** | 79.2 (0.001) | 0.05 | 0.01 |
| 10 | MNIST | $\alpha = 0.2$ | 83.5 (0.01) | **85.9 (0.005)** | 0.1 | 0.05 | **80.4 (0.01)** | 79.0 (0.0005) | 0.05 | 0.01 |
| 11 | | $C = 5$ | 87.5 (0.05) | 88.1 (0.01) | > 0.1 | 0.05 | **82.7 (0.01)** | 80.1 (0.001) | 0.1 | 0.05 |
| 12 | | $C = 2$ | 83.0 (0.1) | **87.3 (0.005)** | > 0.1 | 0.05 | **75.9 (0.01)** | 72.3 (0.001) | 0.1 | 0.005 |
| 13 | | IID | 86.4 (0.05) | 87.0 (0.005) | 0.1 | 0.05 | 77.7 (0.01) | **83.7 (0.001)** | 0.05 | 0.01 |
| 14 | | $\alpha = 5.0$ | 86.1 (0.05) | 87.0 (0.005) | 0.1 | 0.05 | 77.6 (0.005) | **82.7 (0.001)** | 0.05 | 0.005 |
| 15 | | $\alpha = 0.5$ | 84.1 (0.01) | **85.5 (0.005)** | 0.05 | 0.01 | 71.8 (0.01) | **76.9 (0.001)** | 0.05 | 0.005 |
| 16 | CIFAR-10 | $\alpha = 0.2$ | 80.5 (0.01) | **83.5 (0.001)** | 0.05 | 0.01 | **66.9 (0.01)** | 65.0 (0.0005) | 0.05 | 0.001 |
| 17 | | $C = 5$ | 85.5 (0.05) | **86.5 (0.005)** | 0.1 | 0.05 | 74.2 (0.01) | **78.9 (0.001)** | 0.05 | 0.005 |
| 18 | | $C = 2$ | 80.0 (0.05) | **84.7 (0.005)** | 0.1 | 0.05 | **61.6 (0.01)** | 58.7 (0.0005) | 0.05 | 0.001 |

# 6 CONCLUSION

In this work, we first derived the convergence guarantee of SL for non-convex objectives on non-IID data. The results reveal that the convergence of SL is affected by the factors such as data heterogeneity and local update steps. Furthermore, we compare SL against FL theoretically and empirically, ending up with the conclusions that (i) the best and threshold learning rate of SL is smaller than FL; (ii) the performance of SL is worse than FL when the number of local update steps is large on highly non-IID data; (iii) the performance of SL can be better than FL when the number of local update steps is small on highly non-IID data. Our work can bridge the gap between FL and SL, provide deep understanding of these two approaches and guide the deployment of these two in real-world applications.

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

CONTENTS

## A    MORE DETAILS ABOUT SPLIT LEARNING

In this section, we provide more details about SL. More discussions can be found in Gupta & Raskar (2018); Thapa et al. (2021); Duan et al. (2022).

In SL, the complete (or full) ML model is split into two portions. The portion of the complete model maintained by the clients is called *client-side model*. The portion maintained by the server is called *server-side model*. In SL, the client-side model is owned by all the clients, while the server-side model is merely owned by the server. Any client can operate with the server to complete the model training task but can not do it independently (without the help of the server). Then we prepare some basic concepts in the paper.

**Local update (step)**  The process that one client and the server cooperate to conduct the model updates, i.e., the inner loop of Algorithm 1. Note that in SL, though the model update requires communication between the client and server, **the model is still trained on the local dataset**. So the process is called *local update* (Thapa et al., 2020).

**Local model**  The concatenation of the client-side and server-side models after each local update.

**Training round**  The process that all the clients (or a subset of clients selected) complete their local updates, i.e., the outer loop of Algorithm 1. It is also called a *global epoch* or *global round*.

**Global model**  The concatenation of the client-side and server-side models after each global round.

Table 3 summarizes the notations. In particular, **the superscripts and subscripts** of random variables appearing in the paper have the same form — $\mathbf{a}_i^{(r,k)}$, where $r$ is the index of rounds, $k$ is the index of local update steps, $i$ is the index of clients. It means that the random variable $\mathbf{a}$ after the $k$-th local update on the local dataset $\mathcal{D}_i$ in the $r$-th round. The random variable $\mathbf{a}$ can be $\mathbf{x}$ (model parameters), $\xi$ (random samples) or $\mathbf{g}$ (stochastic gradients).

Table 3: Summary of notations appearing in the paper.

| Symbol | Description |
|---|---|
| $T$ | Total number of iterations |
| $R, r$ | Number, index of (training) rounds |
| $N, i$ | Number, index of clients |
| $\mathcal{S}, \|\mathcal{S}\|$ | Subset of clients sampled for training every round and its size |
| $K, k$ | Number, index of local update steps; $K$ represents the maximum local update steps when the number of local update steps varies across clients |
| $n, n_i$ | Sizes of the overall dataset $\mathcal{D}$ and local dataset $\mathcal{D}_i$ at client $i$ |
| $E$ | Local epochs; number of training passes each client makes over its local dataset |
| $b$ | Mini-batch size |
| $\eta_g, \eta_l$ | The global, local learning rate |
| $\alpha, C$ | They are used to denote different distributions by the two mechanisms generating non-IID data respectively (see Section 5.1) |
| $\mathbf{x}^r$ | The complete global model (parameters) in the $r$-th round |
| $\mathbf{x}_i^{(r,k)}$ | The complete local model (parameters) after the $k$-th local update on the local dataset $\mathcal{D}_i$ in the $r$-th round |
| $\mathbf{x}_c, \mathbf{x}_s$ | The client-side model, server-side model |
| $\mathbf{g}_i(\mathbf{x}_i^{(r,k)})$ | Stochastic gradients over on the mini-batch $\xi_i^{(r,k)}$ sampled randomly from $\mathcal{D}_i$, also denotes as $\nabla f_i(\mathbf{x}_i^{(r,k)}; \xi_i^{(r,k)})$ |

**Two modes of SL.** In fact, there are two approaches of training in SL (i) with client-side model synchronization and (ii) without client-side model synchronization (Singh et al., 2019; Duan et al., 2022). In this paper, SL is referred to the first approach, i.e., SL with client-side model synchronization. As shown in Figure 4, there are two modes of SL with client-side model synchronization, including (i) the peer-to-peer mode and (ii) centralized mode. The only difference between the peer-to-peer and centralized mode is the synchronization way of model parameters[4].

- **Peer-to-peer mode.** Clients store the model parameters by themselves after the local updates. So clients get the latest model from other clients.
- **Centralized mode.** Clients send the model parameters to the server after the local updates. So clients get the latest model from the server. The centralized mode requires more communication than peer-to-peer mode.

**Implementation of SL with $\eta_g$ in the peer-to-peer and centralized modes.** In each round, the last client (client $N$ in Figure 4) needs to request for the initial client-side model and conduct the global model update after completing its local update. The server will conduct the global model update synchronously. For the peer-to-peer mode, this process will cause (i) additional communication cost (the red arrow line at the left side of Figure 4) and (ii) additional storage cost of devices and the server (storing the global model). For the centralized mode, this process will cause (i) additional communication cost (the red arrow line at the right side of Figure 4) (ii) additional storage cost of the server (the client-side global model is also stored in the server). The communication cost of SL with global update is given in Table 4 based on Singh et al. (2019).

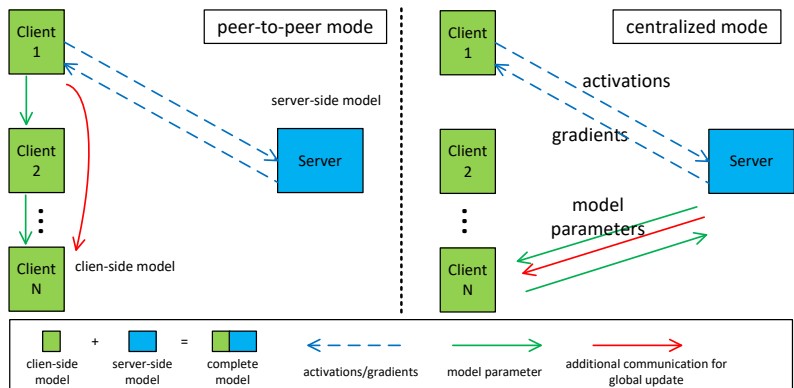

Figure 4: Two modes of SL. The peer-to-peer mode is illustrated in the left side; the centralized mode is illustrated in the right side.

Table 4: Communication cost of FL and SL when running $E$ local epochs for $N$ clients. $\mathbf{x}, \mathbf{x}_c$ are the sizes of the complete model parameters and the client-side model parameters respectively. $n, n_i$ are the sizes of the overall dataset $\mathcal{D}$ and local dataset $\mathcal{D}_i$ respectively. The size of the smashed layer is denoted as $q$. SL with global update need additional communication cost of one client-side model parameter each round.

| Algorithm | Communication of client $i$ | Total communication each round |
|---|---|---|
| FL | $\underbrace{2\mathbf{x}}_{\text{complete model}}$ | $2N\mathbf{x}$ |
| SL (peer-to-peer mode) | $\underbrace{En_iq}_{\text{activations}} + \underbrace{En_iq}_{\text{gradient}} + \underbrace{\mathbf{x}_c}_{\text{client-side model}}$ | $2En + N\mathbf{x}_c$ |
| SL (centralized mode) | $\underbrace{En_iq}_{\text{activations}} + \underbrace{En_iq}_{\text{gradient}} + \underbrace{2\mathbf{x}_c}_{\text{client-side model}}$ | $2En + 2N\mathbf{x}_c$ |
| SL with global update | - | Comm. of vanilla SL $+ \mathbf{x}_c$ |

---

[4]The security and privacy issue is beyond the scope of this work.

# B  ADDITIONAL RELATED WORK

**Variants of SL.** SL is deemed as a promising paradigm for distributed model training at resource-constrained devices, given its computational efficiency on the client side. Most existing works focus on reducing the training delay arising from the relay-based training manner in the multi-user scenario. `SplitFed` (Thapa et al., 2020) is one popular model parallel algorithm that combines the strengths of FL and SL, where each client has one corresponding instance of server-side model in the *main server* to form a pair. Each pair constitutes a complete model and conducts the local update in parallel. After each training round, the *fed server* collects and aggregates on the clien-side local updates. The aggregated client-side model will be disseminated to all the clients before next round. The *main server* does the same operations to the instances of the server-side model. However, in `SplitFed`, the *main server* is required for great computing power to support the multiple instances of server-side model. To address this issue, Thapa et al. (2020) proposed `SplitFedv2`. The only change in `SplitFedv2` is that only one server-side model in *main server*. So the server-side model has to process the activations (or smashed data) of the client-side model in sequence. The model gets updated in every single forward-backward propagation, which means that the server-side model makes more updates than the client-side model. The main point of `SplitFedv3` (Gawali et al., 2021) is substituting alternative client training with alternative the mini-batch training. In vanilla SL, the local model makes one or more training passes over the local dataset (alternative client training). This may cause "catastrophic forgetting" issue (Gawali et al., 2021; Duan et al., 2022). So `SplitFedv3` propose to use alternate mini-batch training, where a client updates its client-side model on one mini-batch, after which the client next in order takes over (Gawali et al., 2021), to mitigate the issue. `SFLG` (Gao et al., 2021) is one generalized variants of `SplitFed`. The clients are allocated to multiple groups. There is one server in each group. The training inside the group is identical to `SplitFedv2`. Then the server-side "global" models per group are aggregated (e.g., weighted averaging) to obtain the server-side global model of all groups. `FedSeq` (Zaccone et al., 2022) is the same as `SFLG` except that the training inside the group is identical to vanilla SL. More variants can be found in Thapa et al. (2021); Duan et al. (2022).

**Convergence analysis of SL.** The convergence analysis of SL is identical to standard Minibatch-SGD on IID data (Wang et al., 2022; Park et al., 2021), so some convergence properties of Minibatch SGD is applied to SL too. The algorithm in Han et al. (2021) reduces the latency and downlink communication on `SplitFed` by adding auxiliary networks at client-side to generate the local loss for model updating. Their convergence analysis combines the analysis of Belilovsky et al. (2020) and `FedAvg`. Wang et al. (2022) proposed `FedLite` to reduce the uplink communication overhead by compressing activations with product quantization and provided the convergence analysis of `FedLite`. However, their convergence recovers that of Minibatch SGD when there is no quantization. SGD with biased gradients (Ajalloeian & Stich, 2020) is also related. However, it only converges to a neighborhood of the solution. Furthermore, we find that Woodworth et al. (2020a;b) compared the convergence of distributed Minibatch SGD and local SGD under homogeneous and heterogeneous settings. To differentiate our work, we show how these algorithms operate in Table 5. For Minibatch SGD, there is no local update and each client computes $K$ stochastic gradients at the same point $\mathbf{x}^r$. FL and SL make $K$ times more updates than Minibatch SGD. However, models in FL are training in parallel, while models in SL are in sequence.

Table 5: The update rules of Minibatch SGD, FL (Local SGD) and SL. We use the descriptions of Local SGD and Minibatch SGD in Woodworth et al. (2020b) and notations in Section 2. $\xi_i^{(r,k)} \sim \mathcal{D}_i$ denotes one random sample from client $i$ for the $k+1$-th local update in the $r$-th round.

| Algorithm | Global update | Local update of client $i$ |
|---|---|---|
| Minibatch SGD | $\mathbf{x}^{r+1} = \mathbf{x}^r - \eta_l \frac{1}{NK} \sum_{i=1}^{N} \sum_{k=0}^{K-1} \nabla f_i(\mathbf{x}^r; \xi_i^{(r,k)})$ | - 
 - |
| Local SGD (or FL) | $\mathbf{x}^{r+1} = \mathbf{x}^r - \eta_l \frac{1}{N} \sum_{i=1}^{N} \sum_{k=0}^{K-1} \nabla f_i(\mathbf{x}_i^{(r,k)}; \xi_i^{(r,k)})$ | Initialize: $\mathbf{x}_i^{(r,0)} = \mathbf{x}^r$ 
 Update: $\mathbf{x}_i^{(r,k+1)} = \mathbf{x}_i^{(r,k)} - \eta_l \nabla f_i(\mathbf{x}_i^{(r,k)}; \xi_i^{(r,k)})$ |
| SL | $\mathbf{x}^{r+1} = \mathbf{x}^r - \eta_l \sum_{i=1}^{N} \sum_{k=0}^{K-1} \nabla f_i(\mathbf{x}_i^{(r,k)}; \xi_i^{(r,k)})$ | Initialize: $\mathbf{x}_i^{(r,0)} = \mathbf{x}_{i-1}^{(r,K)}$ 
 Update: $\mathbf{x}_i^{(r,k+1)} = \mathbf{x}_i^{(r,k)} - \eta_l \nabla f_i(\mathbf{x}_i^{(r,k)}; \xi_i^{(r,k)})$ |

## C  SUMMARY OF THEORIES

There are two methods to give the convergence of SL: (i) bounding the progress of all clients in one round; (ii) bounding the progress of one client in one round. The second method can be given based on the techniques of FL (Khaled et al., 2020; Karimireddy et al., 2020; Wang et al., 2020) directly. However, it only converges to a neighborhood around the stationary point of the global function. This case is similar to the biased SGD (Ajalloeian & Stich, 2020). It is intuitive to get the conclusion since the local stochastic gradient $\nabla f_i(\mathbf{x}; \xi_i)$ ($\xi_i \sim \mathcal{D}_i$) generated by local data of any client $i$ is a biased gradient estimator of the global gradient $\nabla f(\mathbf{x})$ of the global function. For the first method — bounding the progress of all clients in one round, our main contribution, shows that SL can converge to the stationary point of the global function. The main theories are summarized in Table 6.

Table 6: Summary of theories for SL. $\tilde{\eta}$ is the effective learning rate defined in Section 4.3. $\tilde{\eta}$ in "progress of all clients in one round" is defined as $NK\eta_g\eta_l$ while $\tilde{\eta}$ in "progress of one client in one round" is defined as $K\eta_l$. $\eta_g = 1$. $F := f(\mathbf{x}^0) - f(\mathbf{x}^*)$.

| Outline | Theory |
|---|---|
| **Progress of all clients in one round ($\tilde{\eta} = NK\eta_g\eta_l$)** | |
| $\mathbb{E}[\|\nabla f(\mathbf{x}^r)\|^2] \leq \mathcal{O}\left(\frac{f(\mathbf{x}^r) - f(\mathbf{x}^{r+1})}{\tilde{\eta}}\right) + \mathcal{O}\left(\tilde{\eta}^2 G^2 + \frac{\tilde{\eta}^2 \sigma^2}{K}\right) + \mathcal{O}\left(\frac{\tilde{\eta}\sigma^2}{K}\right)$ | Thm. 1 |
| $\mathbb{E}[\|\nabla f(\bar{\mathbf{x}}^R)\|^2] \leq \mathcal{O}\left(\frac{F}{\tilde{\eta}R}\right) + \mathcal{O}\left(\tilde{\eta}^2 G^2 + \frac{\tilde{\eta}^2 \sigma^2}{K}\right) + \mathcal{O}\left(\frac{\tilde{\eta}\sigma^2}{K}\right)$ | Thm. 1 |
| $\mathbb{E}[\|\nabla f(\bar{\mathbf{x}}^R)\|^2] \leq \mathcal{O}\left(\frac{F}{\sqrt{T}}\right) + \mathcal{O}\left(\frac{N^2 K^2 G^2 + N^2 K\sigma^2}{T}\right) + \mathcal{O}\left(\frac{\sigma^2}{\sqrt{T}}\right)$ | Cor. 1 |
| $\mathbb{E}[\|\nabla f(\bar{\mathbf{x}}^R)\|^2] \leq \mathcal{O}\left(\frac{F}{\sqrt{R}}\right) + \mathcal{O}\left(\frac{KG^2 + \sigma^2}{KR}\right) + \mathcal{O}\left(\frac{\sigma^2}{K\sqrt{R}}\right)$ | Cor. 2 |
| $\mathbb{E}[\|\nabla f(\bar{\mathbf{x}}^R)\|^2] \leq \mathcal{O}\left(\frac{F\sqrt{(2B^2+1)}}{R+1}\right) + \mathcal{O}\left(\frac{F^{\frac{2}{3}}(G^2 + \frac{\sigma^2}{K})^{\frac{1}{3}}}{(R+1)^{\frac{2}{3}}}\right) + \mathcal{O}\left(\frac{F\sigma^2}{\sqrt{K(R+1)}}\right)$ | Cor. 3 |
| **Progress of one client in one round ($\tilde{\eta} = K\eta_l$)** | |
| $\mathbb{E}[\|\nabla f(\mathbf{x}_i^{(r,0)})\|^2] \leq \mathcal{O}\left(\frac{f(\mathbf{x}_i^{(r,0)}) - f(\mathbf{x}_{i+1}^{(r,0)})}{\tilde{\eta}}\right) + \mathcal{O}\left(\tilde{\eta}^2 G^2 + \frac{\tilde{\eta}^2 \sigma^2}{K}\right) + \mathcal{O}\left(\frac{\tilde{\eta}\sigma^2}{K}\right) + \mathcal{O}\left(G^2\right)$ | Thm. 3 |
| $\frac{1}{NR}\sum_{r=0}^{R-1}\sum_{i=1}^{N}\mathbb{E}[\|\nabla f(\mathbf{x}_i^{(r,0)})\|^2] \leq \mathcal{O}\left(\frac{F}{\tilde{\eta}NR}\right) + \mathcal{O}\left(\tilde{\eta}^2 G^2 + \frac{\tilde{\eta}^2 \sigma^2}{K}\right) + \mathcal{O}\left(\frac{\tilde{\eta}\sigma^2}{K}\right) + \mathcal{O}\left(G^2\right)$ | Thm. 3 |

## D  PROOF OF RESULTS

### D.1  BASIC TECHNICAL LEMMAS AND NOTATIONS

**Lemma 1.** $\boldsymbol{x}_1, \ldots, \boldsymbol{x}_N$ *are $N$ vectors, then*

$$\|\boldsymbol{x}_i + \boldsymbol{x}_j\|^2 \leq 2\|\boldsymbol{x}_i\|^2 + 2\|\boldsymbol{x}_j\|^2 \tag{7}$$

$$\|\boldsymbol{x}_i + \boldsymbol{x}_j\|^2 \leq (1+a)\|\boldsymbol{x}_i\|^2 + (1+\frac{1}{a})\|\boldsymbol{x}_j\|^2 \quad \textit{for any } a > 0, \tag{8}$$

$$\left\|\sum_{i=1}^{N}\boldsymbol{x}_i\right\|^2 \leq N\sum_{i=1}^{N}\|\boldsymbol{x}_i\|^2. \tag{9}$$

**Lemma 2** (Jensen's inequality). *For any convex function $f$ and any vectors $\boldsymbol{x}_1, \ldots, \boldsymbol{x}_N$ we have*

$$f\left(\frac{1}{N}\sum_{i=1}^{N}\boldsymbol{x}_i\right) \leq \frac{1}{N}\sum_{i=1}^{N}f(\boldsymbol{x}_i). \tag{10}$$

*As a special case with $f(x) = \|x\|^2$, we obtain*

$$\left\|\frac{1}{N}\sum_{i=1}^{N}\boldsymbol{x}_i\right\|^2 \leq \frac{1}{N}\sum_{i=1}^{N}\|\boldsymbol{x}_i\|^2. \tag{11}$$

**Lemma 3.** *Suppose $\{A_k\}_{k=1}^{T}$ is a sequence of random matrices and $\mathbb{E}[A_k|A_{k-1}, A_{k-2}, \ldots, A_1] = \mathbf{0}, \forall k$. Then,*

$$\mathbb{E}\left[\left\|\sum_{k=1}^{T}A_k\right\|_F^2\right] = \sum_{k=1}^{T}\mathbb{E}\left[\|A_k\|_F^2\right]. \tag{12}$$

*Proof.* This is the Lemma 2 of Wang et al. (2020).

$$\mathbb{E}\left[\left\|\sum_{k=1}^{T}A_k\right\|_F^2\right] = \sum_{k=1}^{T}\mathbb{E}\left[\|A_k\|_F^2\right] + \sum_{i=1}^{T}\sum_{j=1, j\neq i}^{T}\mathbb{E}\left[\mathrm{Tr}\{A_i^\top A_j\}\right] \tag{13}$$

$$= \sum_{k=1}^{T}\mathbb{E}\left[\|A_k\|_F^2\right] + \sum_{i=1}^{T}\sum_{j=1, j\neq i}^{T}\mathrm{Tr}\{\mathbb{E}\left[A_i^\top A_j\right] \tag{14}$$

Assume $i < j$. Then, using the law of total expectation,

$$\mathbb{E}\left[A_i^\top A_j\right] = \mathbb{E}\left[A_i^\top \mathbb{E}[A_j|A_i, \ldots, A_1]\right] = \mathbf{0}. \tag{15}$$

$\square$

**Lemma 4** (Bounded Drift). *For any local learning rate satisfying $\eta_l \leq \frac{1}{2NKL}$, the client drift caused by local updates is bounded, as given by:*

$$\sum_{i=1}^{N}\sum_{k=0}^{K-1}\mathbb{E}\left[\left\|\mathbf{x}_i^{(r,k)} - \mathbf{x}^r\right\|^2\right] \leq \frac{2N^3K^2\eta_l^2\sigma^2 + 4N^3K^3\eta_l^2\left(B^2\|\nabla f(\mathbf{x}^r)\|^2 + G^2\right)}{1 - 4N^2K^2\eta_l^2L^2} \tag{16}$$

*Proof.* This proof is based on the proof of Theorem 1: Convergence of Surrogate Objective of Wang et al. (2020). Considering

$$\mathbf{x}_i^{(r,k)} - \mathbf{x}^r = \mathbf{x}_i^{(r,k)} - \mathbf{x}_i^{(r,0)} + \mathbf{x}_i^{(r,0)} - \mathbf{x}_{i-1}^{(r,0)} + \cdots + \mathbf{x}_2^{(r,0)} - \mathbf{x}_1^{(r,0)} \tag{17}$$

$$= -\eta_l\sum_{t=0}^{k-1}\mathbf{g}_i(\mathbf{x}_i^{(r,t)}) - \eta_l\sum_{s=1}^{i-1}\sum_{t=0}^{K-1}\mathbf{g}_s(\mathbf{x}_s^{(r,t)}), \tag{18}$$

we have

$$
\mathbb{E}\left[\left\|\mathbf{x}_i^{(r,k)} - \mathbf{x}^r\right\|^2\right]
$$

$$
= \eta_l^2 \mathbb{E}\left[\left\|\sum_{t=0}^{k-1} \mathbf{g}_i(\mathbf{x}_i^{(r,t)}) + \sum_{s=1}^{i-1} \sum_{t=0}^{K-1} \mathbf{g}_s(\mathbf{x}_s^{(r,t)})\right\|^2\right] \tag{19}
$$

$$
\leq 2\eta_l^2 \mathbb{E}\left[\left\|\sum_{t=0}^{k-1} [\mathbf{g}_i(\mathbf{x}_i^{(r,t)}) - \nabla f_i(\mathbf{x}_i^{(r,t)})] + \sum_{s=1}^{i-1} \sum_{t=0}^{K-1} [\mathbf{g}_s(\mathbf{x}_s^{(r,t)}) - \nabla f_s(\mathbf{x}_s^{(r,t)})]\right\|^2\right]
$$

$$
+ 2\eta_l^2 \mathbb{E}\left[\left\|\sum_{t=0}^{k-1} \nabla f_i(\mathbf{x}_i^{(r,t)}) + \sum_{s=1}^{i-1} \sum_{t=0}^{K-1} \nabla f_s(\mathbf{x}_s^{(r,t)})\right\|^2\right] \tag{20}
$$

$$
\overset{(9)}{\leq} 2i\eta_l^2 \mathbb{E}\left[\left\|\sum_{t=0}^{k-1} [\mathbf{g}_i(\mathbf{x}_i^{(r,t)}) - \nabla f_i(\mathbf{x}_i^{(r,t)})]\right\|^2 + \sum_{s=1}^{i-1} \left\|\sum_{t=0}^{K-1} [\mathbf{g}_s(\mathbf{x}_s^{(r,t)}) - \nabla f_s(\mathbf{x}_s^{(r,t)})]\right\|^2\right]
$$

$$
+ 2i\eta_l^2 \mathbb{E}\left[\left\|\sum_{t=0}^{k-1} \nabla f_i(\mathbf{x}_i^{(r,t)})\right\|^2 + \sum_{s=1}^{i-1} \left\|\sum_{t=0}^{K-1} \nabla f_s(\mathbf{x}_s^{(r,t)})\right\|^2\right] \tag{21}
$$

Applying Lemma 3 to the first term on the right hand side in Eq. (21) and Jensen's Inequality to the second term respectively, we get

$$
\mathbb{E}\left[\left\|\mathbf{x}_i^{(r,k)} - \mathbf{x}^r\right\|^2\right] \leq 2i^2 K \eta_l^2 \sigma^2 + 2iK\eta_l^2 \sum_{s=1}^{i} \sum_{t=0}^{K-1} \mathbb{E}\left[\left\|\nabla f_s(\mathbf{x}_s^{(r,t)})\right\|^2\right] \tag{22}
$$

$$
\overset{(7)}{\leq} 2i^2 K \eta_l^2 \sigma^2 + 4iK\eta_l^2 \sum_{s=1}^{i} \sum_{t=0}^{K-1} \mathbb{E}\left[\|\nabla f_s(\mathbf{x}^r)\|^2\right]
$$

$$
+ 4iK\eta_l^2 \sum_{s=1}^{i} \sum_{t=0}^{K-1} \mathbb{E}\left[\left\|\nabla f_s(\mathbf{x}_s^{(r,t)}) - \nabla f_s(\mathbf{x}^r)\right\|^2\right] \tag{23}
$$

$$
\overset{\text{Asm. }1}{\leq} 2i^2 K \eta_l^2 \sigma^2 + 4iK\eta_l^2 \sum_{s=1}^{i} \sum_{t=0}^{K-1} \mathbb{E}\left[\|\nabla f_s(\mathbf{x}^r)\|^2\right]
$$

$$
+ 4iK\eta_l^2 L^2 \sum_{s=1}^{i} \sum_{t=0}^{K-1} \mathbb{E}\left[\left\|\mathbf{x}_s^{(r,t)} - \mathbf{x}^r\right\|^2\right] \tag{24}
$$

$$
\leq 2i^2 K \eta_l^2 \sigma^2 + 4iK^2 \eta_l^2 \sum_{s=1}^{N} \mathbb{E}\left[\|\nabla f_s(\mathbf{x}^r)\|^2\right]
$$

$$
+ 4iK\eta_l^2 L^2 \underbrace{\sum_{s=1}^{N} \sum_{t=0}^{K-1} \mathbb{E}\left[\left\|\mathbf{x}_s^{(r,t)} - \mathbf{x}^r\right\|^2\right]}_{\mathcal{E}} \tag{25}
$$

Summing up $\mathbb{E}\left[\|\mathbf{x}_i^{(r,k)} - \mathbf{x}^r\|^2\right]$ over $i$ and $k$, we get

$$
\sum_{i=1}^{N} \sum_{k=0}^{K-1} \mathbb{E}\left[\left\|\mathbf{x}_i^{(r,k)} - \mathbf{x}^r\right\|^2\right] \leq 2K^2\eta_l^2\sigma^2 \sum_{i=1}^{N} i^2 + 4K^3\eta_l^2 \sum_{s=1}^{N} \mathbb{E}\left[\|\nabla f_s(\mathbf{x}^r)\|^2\right] \sum_{i=1}^{N} i
$$

$$
+ 4K^2\eta_l^2 L^2 \mathcal{E} \sum_{i=1}^{N} i \tag{26}
$$

$$
\leq 2N^3 K^2 \eta_l^2 \sigma^2 + 4N^2 K^3 \eta_l^2 \sum_{s=1}^{N} \mathbb{E}\left[\|\nabla f_s(\mathbf{x}^r)\|^2\right]
$$

$$
+ 4N^2 K^2 \eta_l^2 L^2 \mathcal{E} \tag{27}
$$

Term $\mathcal{E}$ is equivalent to $\sum_{i=1}^{N} \sum_{k=0}^{K-1} \mathbb{E}[\|\mathbf{x}_i^{(r,k)} - \mathbf{x}^r\|^2]$, so we can rearrange the equation and get:

$$
(1 - 4N^2 K^2 \eta_l^2 L^2)\mathcal{E} \leq 2N^3 K^2 \eta_l^2 \sigma^2 + 4N^2 K^3 \eta_l^2 \sum_{s=1}^{N} \mathbb{E}\left[\|\nabla f_s(\mathbf{x}^r)\|^2\right] \tag{28}
$$

$$
\stackrel{\text{Asm. 3}}{\leq} 2N^3 K^2 \eta_l^2 \sigma^2 + 4N^3 K^3 \eta_l^2 \left(B^2 \|\nabla f(\mathbf{x}^r)\|^2 + G^2\right) \tag{29}
$$

Using $4N^2 K^2 \eta_l^2 L^2 < 1$ and dividing both sides by it yields the claim of Lemma 4. $\qquad\square$

## D.2   PROOF OF THEOREM 1

*Proof.*  Beginning with Assumption 1, we have

$$
f(\mathbf{x}^{r+1}) - f(\mathbf{x}^r) \leq \left\langle \nabla f(\mathbf{x}^r), \mathbf{x}^{r+1} - \mathbf{x}^r \right\rangle + \frac{L}{2} \left\|\mathbf{x}^{r+1} - \mathbf{x}^r\right\|^2. \tag{30}
$$

From Algorithm 1, we know the global model update in round $r$ can be written as:

$$
\mathbf{x}^{r+1} - \mathbf{x}^r = \eta_g(\mathbf{x}_N^{(r,K)} - \mathbf{x}^r) = -\eta_g\eta_l \sum_{i=1}^{N} \sum_{k=0}^{K-1} \mathbf{g}_i(\mathbf{x}_i^{(r,k)}). \tag{31}
$$

For the expectation on $\mathbf{x}^r$, we get

$$
\mathbb{E}[f(\mathbf{x}^{r+1})] - f(\mathbf{x}^r)
$$

$$
\leq \mathbb{E}\left[\left\langle \nabla f(\mathbf{x}^r), -\eta_g\eta_l \sum_{i=1}^{N} \sum_{k=0}^{K-1} \mathbf{g}_i(\mathbf{x}_i^{(r,k)}) \right\rangle\right] + \frac{L}{2}\mathbb{E}\left[\left\|\eta_g\eta_l \sum_{i=1}^{N} \sum_{k=0}^{K-1} \mathbf{g}_i(\mathbf{x}_i^{(r,k)})\right\|^2\right] \tag{32}
$$

$$
= -N\eta_g\eta_l \sum_{k=0}^{K-1} \mathbb{E}\left[\left\langle \nabla f(\mathbf{x}^r), \frac{1}{N} \sum_{i=1}^{N} \nabla f_i(\mathbf{x}_i^{(r,k)}) \right\rangle\right] + \frac{L}{2}\eta_g^2\eta_l^2\mathbb{E}\left[\left\|\sum_{i=1}^{N} \sum_{k=0}^{K-1} \mathbf{g}_i(\mathbf{x}_i^{(r,k)})\right\|^2\right], \tag{33}
$$

where we use $\mathbb{E}[\mathbf{g}_i(\mathbf{x})] = \nabla f_i(\mathbf{x})$ in the equality (see Assumption 2). For the second term on the right hand side (RHS) of Eq. (33), we have:

$$\mathbb{E}\left[\left\|\sum_{i=1}^{N}\sum_{k=0}^{K-1}\mathbf{g}_i(\mathbf{x}_i^{(r,k)})\right\|^2\right] \overset{(7)}{\leq} 2\mathbb{E}\left[\left\|\sum_{i=1}^{N}\sum_{k=0}^{K-1}\left[\mathbf{g}_i(\mathbf{x}_i^{(r,k)}) - \nabla f_i(\mathbf{x}_i^{(r,k)})\right]\right\|^2\right]$$

$$+ 2\mathbb{E}\left[\left\|\sum_{i=1}^{N}\sum_{k=0}^{K-1}\nabla f_i(\mathbf{x}_i^{(r,k)})\right\|^2\right] \tag{34}$$

$$\overset{(9),\text{Lem. }3}{\leq} 2\mathbb{E}\left[N\sum_{i=1}^{N}\sum_{k=0}^{K-1}\left\|\mathbf{g}_i(\mathbf{x}_i^{(r,k)}) - \nabla f_i(\mathbf{x}_i^{(r,k)})\right\|^2\right]$$

$$+ 2\mathbb{E}\left[\left\|\sum_{i=1}^{N}\sum_{k=0}^{K-1}\nabla f_i(\mathbf{x}_i^{(r,k)})\right\|^2\right] \tag{35}$$

$$\overset{\text{Asm. }2}{\leq} 2N^2K\sigma^2 + 2\mathbb{E}\left[\left\|\sum_{i=1}^{N}\sum_{k=0}^{K-1}\nabla f_i(\mathbf{x}_i^{(r,k)})\right\|^2\right]. \tag{36}$$

Note in Eq. (35) that we apply Jensen's Inequality first before Lemma 3 to the first term of RHS, since the data across clients are non-IID. However, if the data across clients are IID, we can get a tighter bound of $NK\sigma^2$. Then plugging Eq. (36) into Eq. (33), we have:

$$\mathbb{E}[f(\mathbf{x}^{r+1})] - f(\mathbf{x}^r)$$

$$\leq -N\eta_g\eta_l\sum_{k=0}^{K-1}\mathbb{E}\left[\left\langle\nabla f(\mathbf{x}^r), \frac{1}{N}\sum_{i=1}^{N}\nabla f_i(\mathbf{x}_i^{(r,k)})\right\rangle\right] + L\eta_g^2\eta_l^2\mathbb{E}\left[\left\|\sum_{i=1}^{N}\sum_{k=0}^{K-1}\nabla f_i(\mathbf{x}_i^{(r,k)})\right\|^2\right]$$

$$+ N^2KL\eta_g^2\eta_l^2\sigma^2 \tag{37}$$

$$= -\frac{N\eta_g\eta_l}{2}\sum_{k=0}^{K-1}\left[\|\nabla f(\mathbf{x}^r)\|^2 + \mathbb{E}\left\|\frac{1}{N}\sum_{i=1}^{N}\nabla f_i(\mathbf{x}_i^{(r,k)})\right\|^2 - \mathbb{E}\left\|\frac{1}{N}\sum_{i=1}^{N}\nabla f_i(\mathbf{x}_i^{(r,k)}) - \nabla f(\mathbf{x}^r)\right\|^2\right]$$

$$+ L\eta_g^2\eta_l^2\mathbb{E}\left[\left\|\sum_{i=1}^{N}\sum_{k=0}^{K-1}\nabla f_i(\mathbf{x}_i^{(r,k)})\right\|^2\right] + N^2KL\eta_g^2\eta_l^2\sigma^2, \tag{38}$$

where we use the fact that $2\langle a, b\rangle = \|a\|^2 + \|b\|^2 - \|a-b\|^2$ in the last equation. Note that

$$-\frac{1}{2}N\eta_g\eta_l\sum_{k=0}^{K-1}\mathbb{E}\left[\left\|\frac{1}{N}\sum_{i=1}^{N}\nabla f_i(\mathbf{x}_i^{(r,k)})\right\|^2\right] + L\eta_g^2\eta_l^2\mathbb{E}\left[\left\|\sum_{i=1}^{N}\sum_{k=0}^{K-1}\nabla f_i(\mathbf{x}_i^{(r,k)})\right\|^2\right]$$

$$\overset{(9)}{\leq} -\frac{1}{2}\frac{1}{N}\eta_g\eta_l\sum_{k=0}^{K-1}\mathbb{E}\left[\left\|\sum_{i=1}^{N}\nabla f_i(\mathbf{x}_i^{(r,k)})\right\|^2\right] + L\eta_g^2\eta_l^2K\sum_{k=0}^{K-1}\mathbb{E}\left[\left\|\sum_{i=1}^{N}\nabla f_i(\mathbf{x}_i^{(r,k)})\right\|^2\right]$$

$$= -\frac{1}{2N}\eta_g\eta_l(1 - 2NKL\eta_g\eta_l)\sum_{k=0}^{K-1}\mathbb{E}\left[\left\|\sum_{i=1}^{N}\nabla f_i(\mathbf{x}_i^{(r,k)})\right\|^2\right] \tag{39}$$

and $\quad \mathbb{E}\left[\left\|\frac{1}{N}\sum_{i=1}^{N}\nabla f_i(\mathbf{x}_i^{(r,k)}) - \nabla f(\mathbf{x}^r)\right\|^2\right] = \mathbb{E}\left[\left\|\frac{1}{N}\sum_{i=1}^{N}[\nabla f_i(\mathbf{x}_i^{(r,k)}) - \nabla f_i(\mathbf{x}^r)]\right\|^2\right]$

$$\overset{(11)}{\leq} \frac{1}{N}\sum_{i=1}^{N}\mathbb{E}\left[\left\|\nabla f_i(\mathbf{x}_i^{(r,k)}) - \nabla f_i(\mathbf{x}^r)\right\|^2\right]$$

$$\overset{\text{Asm. }1}{\leq} \frac{1}{N}\sum_{i=1}^{N}L^2\mathbb{E}\left[\left\|\mathbf{x}_i^{(r,k)} - \mathbf{x}^r\right\|^2\right], \tag{40}$$

where the first equality of Eq. (40) results from the fact that $\nabla f(\mathbf{x}^r) = \frac{1}{N} \sum_{i=1}^{N} \nabla f_i(\mathbf{x}^r)$. By plugging Eq. (39) and Eq. (40) into Eq. (38) and using $\mathbf{2NK\eta_g\eta_l L \leq 1}$, we get

$$\mathbb{E}[f(\mathbf{x}^{r+1})] - f(\mathbf{x}^r) \leq -\frac{1}{2} NK\eta_g\eta_l \|\nabla f(\mathbf{x}^r)\|^2 + \frac{1}{2} L^2 \eta_g\eta_l \sum_{i=1}^{N}\sum_{k=0}^{K-1} \mathbb{E}\left[\left\|\mathbf{x}_i^{(r,k)} - \mathbf{x}^r\right\|^2\right]$$
$$+ N^2 KL\eta_g^2\eta_l^2\sigma^2. \tag{41}$$

Then we use Lemma 4:

$$\frac{\mathbb{E}[f(\mathbf{x}^{r+1})] - f(\mathbf{x}^r)}{NK\eta_g\eta_l} \leq -\frac{1}{2}\left(1 - \frac{DB^2}{1-D}\right) \|\nabla f(\mathbf{x}^r)\|^2 + NL\eta_g\eta_l\sigma^2$$
$$+ \frac{1}{1-D}\left(N^2 KL^2\eta_l^2\sigma^2 + 2N^2K^2L^2\eta_l^2 G^2\right), \tag{42}$$

where $D = 4N^2K^2L^2\eta_l^2$. Then using $\mathbf{D \leq \frac{1}{2B^2+1}}$ and $B \geq 1$, we get

$$\frac{\mathbb{E}[f(\mathbf{x}^{r+1})] - f(\mathbf{x}^r)}{NK\eta_g\eta_l} \leq -\frac{1}{4}\|\nabla f(\mathbf{x}^r)\|^2 + NL\eta_g\eta_l\sigma^2$$
$$+ (1 + \frac{1}{2B^2})\left(N^2KL^2\eta_l^2\sigma^2 + 2N^2K^2L^2\eta_l^2G^2\right) \tag{43}$$
$$\leq -\frac{1}{4}\|\nabla f(\mathbf{x}^r)\|^2 + NL\eta_g\eta_l\sigma^2 + \frac{3}{2}N^2KL^2\eta_l^2\sigma^2 + 3N^2K^2L^2\eta_l^2G^2 \tag{44}$$

Taking unconditional expectation, rearranging the terms and then averaging the above equation over $r = \{0, \cdots, R-1\}$, we have

$$\frac{1}{R}\sum_{r=0}^{R-1}\mathbb{E}\|\nabla f(\mathbf{x}^r)\|^2 \leq \frac{4[f(\mathbf{x}^0) - f(\mathbf{x}^*)]}{NK\eta_g\eta_l R} + 12N^2K^2\eta_l^2L^2G^2 + 6N^2K\eta_l^2L^2\sigma^2 + 4N\eta_g\eta_l L\sigma^2$$

Using the fact that $\mathbb{E}\left\|\nabla f(\bar{\mathbf{x}}^R)\right\|^2 \leq \frac{1}{R}\sum_{r=0}^{R-1}\mathbb{E}\|\nabla f(\mathbf{x}^r)\|^2$ where $\bar{\mathbf{x}}^R = \frac{1}{R}\sum_{r=0}^{R-1}\mathbf{x}^r$, we get the Eq. (5). Finally, we summarize the constraints:

$$D = 4N^2K^2L^2\eta_l^2 \leq \frac{1}{2B^2+1} \tag{45}$$
$$2NK\eta_g\eta_l L \leq 1 \tag{46}$$
$$2NKL\eta_l \leq 1, \tag{47}$$

where the last inequality is from Lemma 4. The overall constraint is given as:

$$\eta_l \leq \frac{1}{2NKL}\min\left\{\frac{1}{\sqrt{2B^2+1}}, \frac{1}{\eta_g}\right\} \tag{48}$$

Now we complete the proof of Theorem 1. □

**Corollary 2.** *Choose* $NK\eta_g\eta_l = \frac{1}{\sqrt{R}}$ *and apply the result of Theorem 1. For sufficiently large R, it holds that*

$$\mathbb{E}[\|\nabla f(\bar{\mathbf{x}}^R)\|^2] \leq \mathcal{O}\left(\frac{F}{\sqrt{R}}\right) + \mathcal{O}\left(\frac{KG^2 + \sigma^2}{\sqrt{KR}}\right) + \mathcal{O}\left(\frac{\sigma^2}{K\sqrt{R}}\right), \tag{49}$$

*where* $F := f(\mathbf{x}^0) - f(\mathbf{x}^*)$, $R$ *is the total rounds,* $\bar{\mathbf{x}}^R = \frac{1}{R}\sum_{r=0}^{R-1}\mathbf{x}^r$ *is the averaged global model over the R rounds.*

**Corollary 3.** *Apply the result of Theorem 1. There exits* $\eta_l$, *such that*

$$\mathbb{E}[\|\nabla f(\bar{\mathbf{x}}^R)\|^2] \leq \mathcal{O}\left(\frac{F\sqrt{(2B^2+1)}}{R+1}\right) + \mathcal{O}\left(\frac{F^{\frac{2}{3}}(G^2 + \frac{\sigma^2}{K})^{\frac{1}{3}}}{(R+1)^{\frac{2}{3}}}\right) + \mathcal{O}\left(\frac{F\sigma^2}{\sqrt{K(R+1)}}\right), \tag{50}$$

*where* $F := f(\mathbf{x}^0) - f(\mathbf{x}^*)$, $R$ *is the total rounds,* $\bar{\mathbf{x}}^R = \frac{1}{R}\sum_{r=0}^{R-1}\mathbf{x}^r$ *is the averaged global model over the R rounds.*

*Proof.* Applying Lemma 2 (sub-linear convergence rate) of Karimireddy et al. (2020) to Eq. (44), we get the the claim of this corollary. □

### D.3 PROOF OF THEOREM 2

**Theorem 2.** *Let Assumptions 1, 2 and 3 hold. Suppose that the local learning rate satisfies $\eta_l \leq \frac{1}{2KL} \min \left\{ \frac{1}{\sqrt{2B^2+1}}, \frac{1}{\eta_g} \right\}$. For Algorithm 1, it holds that*

$$\mathbb{E}[\|\nabla f(\bar{\mathbf{x}}^R)\|^2] \leq \underbrace{\frac{4[f(\mathbf{x}^0) - f(\mathbf{x}^*)]}{K\eta_g\eta_l R}}_{T_1:initialization\ error} + \underbrace{12K(K-1)\eta_l^2 L^2 G^2 + 6(K-1)\eta_l^2 L^2 \sigma^2}_{T_2:client\ drift\ error} + \underbrace{\frac{4\eta_g\eta_l L\sigma^2}{N}}_{T_3:global\ variance} ,$$
(51)

*where $\bar{\mathbf{x}}^R = \frac{1}{R}\sum_{r=0}^{R-1} \mathbf{x}^r$ is the averaged global model over the R rounds.*

*Proof.* This is almost the same as Theorem 1 of Wang et al. (2020), we reproduce here for convenience of comparison between FL and SL. The proof is similar to that of Theorem 1. □

### D.4 PROOF OF THEOREM 3

Here we use Assumption 4 to replace Assumption 3, one stronger assumption (than Assumption 3) used in Lian et al. (2017).

**Assumption 4.** *There exist constants $G \geq 0$ such that*

$$\mathbb{E}_{i\sim\mathcal{U}([N])} \left[\|\nabla f_i(\mathbf{x}) - \nabla f(\mathbf{x})\|\right] \leq G^2,$$
(52)

*where $i$ is uniformly sampled from $\{1, \ldots, N\}$. In the IID case, $G = 0$.*

**Theorem 3** (Progress of one client in one round). *Let Assumptions 1, 2 and 3 hold. Suppose that the local learning rate satisfies $\eta_l \leq \frac{1}{2\sqrt{5}KL}$. For Algorithm 1, it holds that*

$$\frac{1}{NR} \sum_{r=0}^{R-1} \sum_{i=1}^{N} \mathbb{E}\left[\left\|\nabla f(\mathbf{x}_i^{(r,0)})\right\|^2\right] \leq \frac{4[f(\mathbf{x}^0) - f(\mathbf{x}^*)]}{NK\eta_l R} + 40K^2 L^2 \eta_l^2 G^2$$
$$+ 10KL^2\eta_l^2\sigma^2 + 4L\eta_l\sigma^2 + 4G^2,$$
(53)

*where $\bar{\mathbf{x}}^R = \frac{1}{R}\sum_{r=0}^{R-1} \mathbf{x}^r$ is the averaged global model over the R rounds.*

*Proof.* Different from Theorem 1, we bound the progeress of one client in one round. Beginning with Assumption 1, we have:

$$\mathbb{E}\left[f(\mathbf{x}_{i+1}^{(r,0)}) - f(\mathbf{x}_i^{(r,0)})\right] \leq \mathbb{E}\left[\left\langle\nabla f(\mathbf{x}_i^{(r,0)}), \mathbf{x}_{i+1}^{(r,0)} - \mathbf{x}_i^{(r,0)}\right\rangle\right] + \frac{L}{2}\mathbb{E}\left[\left\|\mathbf{x}_{i+1}^{(r,0)} - \mathbf{x}_i^{(r,0)}\right\|^2\right]$$
(54)

From Algorithm 1, we know the local update of client $i$ in round $r$ can be written as:

$$\mathbf{x}_{i+1}^{(r,0)} - \mathbf{x}_i^{(r,0)} = -\eta_l \sum_{k=0}^{K-1} \mathbf{g}_i(\mathbf{x}_i^{(r,k)}).$$
(55)

For the expectation on $\mathbf{x}_i^{(r,0)}$, we get

$$\mathbb{E}\left[f(\mathbf{x}_{i+1}^{(r,0)}) - f(\mathbf{x}_i^{(r,0)})\right]$$

$$\leq \mathbb{E}\left[\left\langle\nabla f(\mathbf{x}_i^{(r,0)}), -\eta_l \sum_{k=0}^{K-1} \mathbf{g}_i(\mathbf{x}_i^{(r,k)})\right\rangle\right] + \frac{L}{2}\mathbb{E}\left[\left\|\eta_l \sum_{k=0}^{K-1} \mathbf{g}_i(\mathbf{x}_i^{(r,k)})\right\|^2\right]$$
(56)

$$= -\eta_l \sum_{k=0}^{K-1} \mathbb{E}\left[\left\langle\nabla f(\mathbf{x}_i^{(r,0)}), \nabla f_i(\mathbf{x}_i^{(r,k)})\right\rangle\right] + \frac{L}{2}\eta_l^2 \mathbb{E}\left[\left\|\sum_{k=0}^{K-1} \mathbf{g}_i(\mathbf{x}_i^{(r,k)})\right\|^2\right],$$
(57)

where we use $\mathbb{E}[\mathbf{g}_i(\mathbf{x})] = \nabla f_i(\mathbf{x})$ in the equality (see Assumption 2). For the second term on the right hand side (RHS) of Eq. (57), we have:

$$\mathbb{E}\left[\left\|\sum_{k=0}^{K-1} \mathbf{g}_i(\mathbf{x}_i^{(r,k)})\right\|^2\right] \stackrel{(7)}{\leq} 2\mathbb{E}\left[\left\|\sum_{k=0}^{K-1}\left[\mathbf{g}_i(\mathbf{x}_i^{(r,k)}) - \nabla f_i(\mathbf{x}_i^{(r,k)})\right]\right\|^2\right]$$

$$+ 2\mathbb{E}\left[\left\|\sum_{k=0}^{K-1} \nabla f_i(\mathbf{x}_i^{(r,k)})\right\|^2\right] \tag{58}$$

$$\stackrel{\text{Lem. }3}{\leq} 2\mathbb{E}\left[\sum_{k=0}^{K-1}\left\|\mathbf{g}_i(\mathbf{x}_i^{(r,k)}) - \nabla f_i(\mathbf{x}_i^{(r,k)})\right\|^2\right]$$

$$+ 2\mathbb{E}\left[\left\|\sum_{k=0}^{K-1} \nabla f_i(\mathbf{x}_i^{(r,k)})\right\|^2\right] \tag{59}$$

$$\stackrel{\text{Asm. }2}{\leq} 2K\sigma^2 + 2\mathbb{E}\left[\left\|\sum_{k=0}^{K-1} \nabla f_i(\mathbf{x}_i^{(r,k)})\right\|^2\right]. \tag{60}$$

Then plugging Eq. (60) into Eq. (57), we have:

$$\mathbb{E}\left[f(\mathbf{x}_{i+1}^{(r,0)}) - f(\mathbf{x}_i^{(r,0)})\right]$$

$$\leq -\eta_l \sum_{k=0}^{K-1} \mathbb{E}\left[\left\langle \nabla f(\mathbf{x}_i^{(r,0)}), \nabla f_i(\mathbf{x}_i^{(r,k)})\right\rangle\right] + L\eta_l^2 \mathbb{E}\left[\left\|\sum_{k=0}^{K-1} \nabla f_i(\mathbf{x}_i^{(r,k)})\right\|^2\right] + KL\eta_l^2\sigma^2 \tag{61}$$

$$= -\frac{\eta_l}{2} \sum_{k=0}^{K-1}\left[\left\|\nabla f(\mathbf{x}_i^{(r,0)})\right\|^2 + \mathbb{E}\left\|\nabla f_i(\mathbf{x}_i^{(r,k)})\right\|^2 - \mathbb{E}\left\|\nabla f_i(\mathbf{x}_i^{(r,k)}) - \nabla f(\mathbf{x}_i^{(r,0)})\right\|^2\right]$$

$$+ L\eta_l^2 \mathbb{E}\left[\left\|\sum_{k=0}^{K-1} \nabla f_i(\mathbf{x}_i^{(r,k)})\right\|^2\right] + KL\eta_l^2\sigma^2, \tag{62}$$

where we use the fact that $2\langle a, b\rangle = \|a\|^2 + \|b\|^2 - \|a-b\|^2$ in the last equation. Note that

$$-\frac{1}{2}\eta_l \sum_{k=0}^{K-1} \mathbb{E}\left[\left\|\nabla f_i(\mathbf{x}_i^{(r,k)})\right\|^2\right] + L\eta_l^2 \mathbb{E}\left[\left\|\sum_{k=0}^{K-1} \nabla f_i(\mathbf{x}_i^{(r,k)})\right\|^2\right]$$

$$\stackrel{(9)}{\leq} -\frac{1}{2}\eta_l \sum_{k=0}^{K-1} \mathbb{E}\left[\left\|\nabla f_i(\mathbf{x}_i^{(r,k)})\right\|^2\right] + L\eta_l^2 K \sum_{k=0}^{K-1} \mathbb{E}\left[\left\|\nabla f_i(\mathbf{x}_i^{(r,k)})\right\|^2\right]$$

$$= -\frac{1}{2}\eta_l(1 - 2KL\eta_l) \sum_{k=0}^{K-1} \mathbb{E}\left[\left\|\nabla f_i(\mathbf{x}_i^{(r,k)})\right\|^2\right] \tag{63}$$

$$\text{and} \quad \mathbb{E}\left[\left\|\nabla f_i(\mathbf{x}_i^{(r,k)}) - \nabla f(\mathbf{x}_i^{(r,0)})\right\|^2\right] = 2\mathbb{E}\left[\left\|\nabla f_i(\mathbf{x}_i^{(r,k)}) - \nabla f_i(\mathbf{x}_i^{(r,0)})\right\|^2\right]$$

$$+ 2\mathbb{E}\left[\left\|\nabla f_i(\mathbf{x}_i^{(r,0)}) - \nabla f(\mathbf{x}_i^{(r,0)})\right\|^2\right] \tag{64}$$

$$\stackrel{\text{Asm. }1}{\leq} 2L^2 \mathbb{E}\left[\left\|\mathbf{x}_i^{(r,k)} - \mathbf{x}_i^{(r,0)}\right\|^2\right] + 2G^2. \tag{65}$$

By plugging Eq. (63) and Eq. (65) into Eq. (62) and using $\mathbf{2K\eta_l L \leq 1}$, we get

$$\mathbb{E}\left[f(\mathbf{x}_{i+1}^{(r,0)}) - f(\mathbf{x}_i^{(r,0)})\right] \leq -\frac{1}{2}K\eta_l \left\|\nabla f(\mathbf{x}_i^{(r,0)})\right\|^2 + L^2\eta_l \sum_{k=0}^{K-1} \mathbb{E}\left[\left\|\mathbf{x}_i^{(r,k)} - \mathbf{x}_i^{(r,0)}\right\|^2\right]$$

$$+ K\eta_l G^2 + KL\eta_l^2\sigma^2. \tag{66}$$

Then using the bounded client-drift in Wang et al. (2020), i.e.,

$$\sum_{k=0}^{K-1} \mathbb{E}\left[\left\|\mathbf{x}_i^{(r,k)} - \mathbf{x}_i^{(r,0)}\right\|^2\right] \le \frac{4K^3\eta_l^2 \left\|\nabla f_i(\mathbf{x}_i^{(r,0)})\right\|^2 + 2K^2\eta_l^2\sigma^2}{1 - 4K^2L^2\eta_l^2}, \tag{67}$$

we can get

$$\frac{\mathbb{E}\left[f(\mathbf{x}_{i+1}^{(r,0)}) - f(\mathbf{x}_i^{(r,0)})\right]}{K\eta_l} \le -\frac{1}{2}\left(1 - \frac{2D}{1-D}\right)\left\|\nabla f(\mathbf{x}_i^{(r,0)})\right\|^2 + L\eta_l\sigma^2 + G^2$$
$$+ \frac{2D}{1-D}G^2 + \frac{2KL^2\eta_l^2\sigma^2}{1-D}, \tag{68}$$

where $D = 4K^2L^2\eta_l^2$. Then using $\boldsymbol{D} \le \frac{1}{5}$ ($\frac{1}{1-D} \le \frac{5}{4}$) and $B \ge 1$, we get

$$\frac{\mathbb{E}\left[f(\mathbf{x}_{i+1}^{(r,0)}) - f(\mathbf{x}_i^{(r,0)})\right]}{K\eta_l} \le -\frac{1}{4}\left\|\nabla f(\mathbf{x}_i^{(r,0)})\right\|^2 + L\eta_l\sigma^2 + G^2$$
$$+ \frac{5}{2}KL^2\eta_l^2\sigma^2 + 10K^2L^2\eta_l^2G^2 \tag{69}$$

Taking unconditional expectation, rearranging the terms and then averaging the above equation over $i = \{1, \cdots, N\}, r = \{0, \cdots, R-1\}$, we have

$$\frac{1}{NR}\sum_{r=0}^{R-1}\sum_{i=1}^{N}\mathbb{E}\left[\left\|\nabla f(\mathbf{x}_i^{(r,0)})\right\|^2\right] \le \frac{4[f(\mathbf{x}^0) - f(\mathbf{x}^*)]}{NK\eta_l R} + 40K^2L^2\eta_l^2G^2$$
$$+ 10KL^2\eta_l^2\sigma^2 + 4L\eta_l\sigma^2 + 4G^2 \tag{70}$$

Using the fact that $\mathbb{E}\left\|\nabla f(\bar{\mathbf{x}}^R)\right\|^2 \le \frac{1}{R}\sum_{r=0}^{R-1}\mathbb{E}\left\|\nabla f(\mathbf{x}^r)\right\|^2$ where $\bar{\mathbf{x}}^R = \frac{1}{R}\sum_{r=0}^{R-1}\mathbf{x}^r$, we get the Eq. (53). Finally, we summarize the constraints:

$$D = 4K^2L^2\eta_l^2 \le \frac{1}{5} \tag{71}$$

$$2K\eta_l L \le 1. \tag{72}$$

The overall constraint is given as:

$$\eta_l \le \frac{1}{2\sqrt{5}KL} \tag{73}$$

Now we complete the proof of Theorem 1. $\qquad\square$

## D.5 EXTREME CASES

**Theorem 1 recovers the convergence of SGD when $N = 1$ and $K = 1$.** Let us focus on the proof of Lemma 4. When $N = 1$ and $K = 1$, the client drift will reduce to:

$$\sum_{i=1}^{N}\sum_{k=0}^{K-1}\mathbb{E}\left[\left\|\mathbf{x}_i^{(r,k)} - \mathbf{x}^r\right\|^2\right] = \mathbb{E}\left[\left\|\mathbf{x}_1^{(r,0)} - \mathbf{x}^r\right\|^2\right] = \mathbf{0}, \tag{74}$$

where $\mathbf{x}_1^{(r,0)} = \mathbf{x}^r$ (see Algorithm 1). Thus the client drift error of Eq. (5) will be removed, which recovers the result of SGD (Bottou et al., 2018).

# E MORE EXPERIMENTAL DETAILS

**Platform.** We train LeNet-5 on MNIST and Fashion-MNIST with Nvidia GeForce RTX 3070 Ti, VGG-11 on CIFAR-10 with Nvidia 3090 Ti. The algorithms are implemented by PyTorch. We use the random seed "1234" by default. We use vanilla SGD algorithm with momentem = 0.9 and weight decay = 1e-4 as He et al. (2020). The detailed information of the models and other information can be found in our code.

## E.1 MORE RESULTS OF SL

This section is complimentary to Section 5.2 to study the factors that affects the performance of SL. We report the details of the experiments, such as $\eta_l, \eta_g, b$.

**Effect of data heterogeneity.**

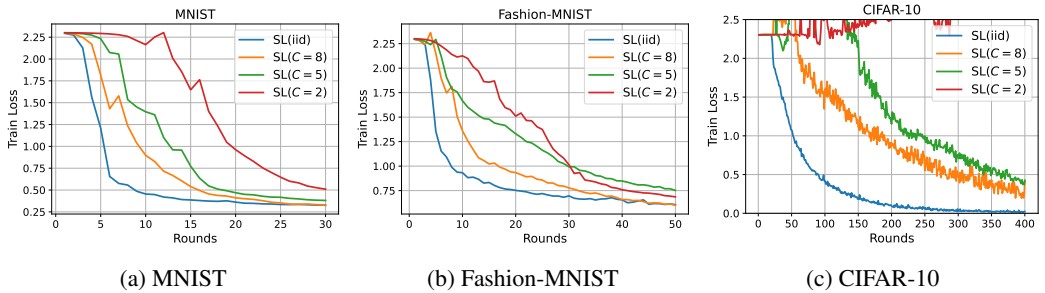

|              (a) MNIST              |        (b) Fashion-MNIST        |        (c) CIFAR-10        |

Figure 5: Effect of data heterogeneity. (a) MNIST, $N = 10$, $b = 1000$, $\eta_l = 0.01$, $\eta_g = 1.0$; (b) Fashion-MNIST, $N = 10$, $b = 1000$, $\eta_l = 0.01$, $\eta_g = 1.0$; (c) CIFAR-10, $N = 10$, $b = 100$, $\eta_l = 0.001$, $\eta_g = 1.0$.

**Effect of $K$.**

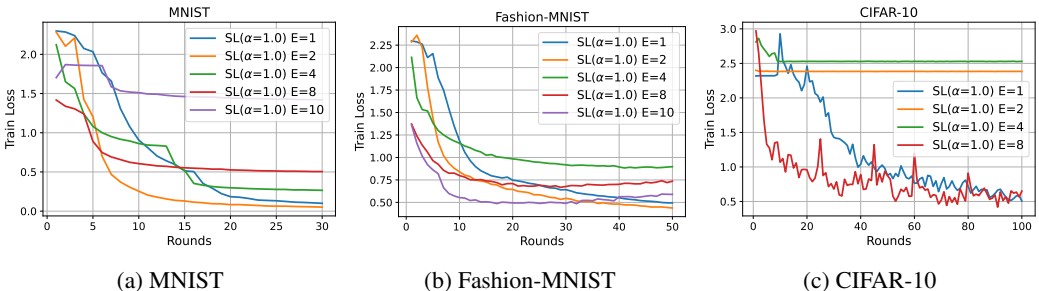

|              (a) MNIST              |        (b) Fashion-MNIST        |        (c) CIFAR-10        |

Figure 6: Effect of $K$. $\text{Dir}_{10}(1.0)$ is used. (a) MNIST, $N = 10$, $b = 1000$, $\eta_l = 0.01$, $\eta_g = 1.0$; (b) Fashion-MNIST, $N = 10$, $b = 1000$, $\eta_l = 0.01$, $\eta_g = 1.0$; (c) CIFAR-10, $N = 10$, $b = 100$, $\eta_l = 0.005$, $\eta_g = 1.0$.

**Effect of $\eta_g$.** The experimental results on Fashion-MNIST and CIFAR-10 are shown in Figure 7. We can see that $\eta_g$ can be helpful in some cases, especially when $\eta_l$ is small. However, there is still a big gap between theory and practice. Further research is required.

## E.2 MORE COMPARISONS BETWEEN FL AND SL IN CROSS-DEVICE SETTING

The learning rates of FL and SL are selected from $\{0.0005, 0.001, 0.005, 0.01, 0.05, 0.1\}$. We report the overall results of comparisons between FL and SL on MNIST, Fashion-MNIST and CIFAR-10 datasets with different learning rates in Table 7, Table 8 and Table 9 respectively. Table 2 in the main body are based on these three tables here. "L-M", "L-M" and "V-10" denote LeNet-5 on MNIST, LeNet-5 on Fashion-MNIST and VGG-11 on CIFAR-10 respectively. We highlight the "best" test accuracy among all chosen learning rates with blue for FL and red for SL. We underline the test accuracy of the "threshold" learning rate with blue for FL and red for SL.

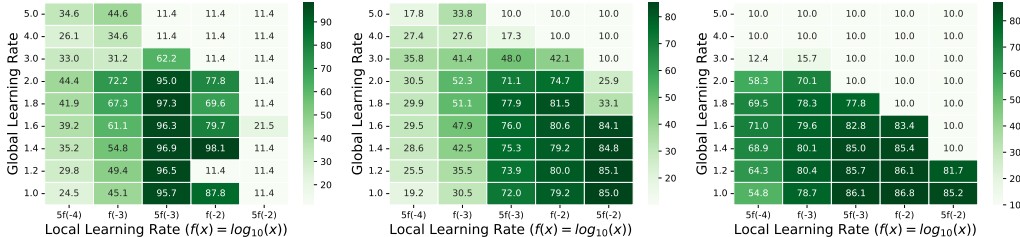

Figure 7: Test accuracy for various local/global learning rates combination. $\text{Dir}_{10}(10.0)$ is used. For MNIST, $N = 10$, $b = 1000$, $E = 1$; We average test accuracy over the last 10 rounds from 30 total rounds; (b) For Fashion-MNIST, $N = 10$, $b = 1000$, $E = 1$; We average test accuracy over the last 10 rounds from 30 total rounds; (c) For CIFAR-10, $N = 10$, $b = 100$, $E = 1$. We average test accuracy over the last 20 rounds from 100 total rounds.

Table 7: The detailed results of FL and SL with different learning rates on MNIST dataset. We average the test accuracy over the last 100 rounds from 1000 total rounds when $E = 1$; average the test accuracy over the last 10 rounds from 100 total rounds when $E = 10$.

| Tag | $N$ | Dist. | $E$ | $b$ | 0.0005 FL | 0.0005 SL | 0.001 FL | 0.001 SL | 0.005 FL | 0.005 SL | 0.01 FL | 0.01 SL | 0.05 FL | 0.05 SL | 0.1 FL | 0.1 SL |
|-----|-----|-------|-----|-----|------|------|------|------|------|------|------|------|------|------|------|------|
| L-M | 1000 | IID | 1 | 10 | 67.8 | 97.8 | 92.0 | 98.4 | 98.3 | 98.9 | 98.7 | 89.2 | 99.0 | 11.3 | 99.0 | 11.3 |
| L-M | 1000 | $\alpha = 10.0$ | 1 | 10 | 78.1 | 98.3 | 93.2 | 98.7 | 98.5 | 99.0 | 98.7 | 98.9 | 99.0 | 11.3 | 98.7 | 11.3 |
| L-M | 1000 | $\alpha = 5.0$ | 1 | 10 | 80.2 | 97.9 | 93.4 | 98.7 | 98.5 | 98.9 | 98.8 | 98.9 | 99.1 | 11.3 | 98.8 | 11.3 |
| L-M | 1000 | $\alpha = 0.5$ | 1 | 10 | 81.3 | 97.9 | 93.6 | 98.6 | 98.4 | 98.8 | 98.8 | 98.7 | 98.9 | 11.3 | 98.3 | 11.3 |
| L-M | 1000 | $\alpha = 0.2$ | 1 | 10 | 85.9 | 98.3 | 94.0 | 98.6 | 98.3 | 98.8 | 98.7 | 98.7 | 98.7 | 11.3 | 11.3 | 11.3 |
| L-M | 1000 | $C = 5$ | 1 | 10 | 65.3 | 98.3 | 91.6 | 98.6 | 98.3 | 98.9 | 98.7 | 98.9 | 99.0 | 11.3 | 98.9 | 11.3 |
| L-M | 1000 | $C = 2$ | 1 | 10 | 56.4 | 98.2 | 89.2 | 98.6 | 97.9 | 99.0 | 98.5 | 99.0 | 98.8 | 11.3 | 98.8 | 11.3 |
| L-M | 1000 | IID | 10 | 10 | 93.4 | 97.3 | 95.6 | 87.9 | 97.6 | 97.6 | 97.9 | 96.2 | 95.5 | 11.3 | 11.3 | 11.3 |
| L-M | 1000 | $\alpha = 10.0$ | 10 | 10 | 93.8 | 97.5 | 96.2 | 97.5 | 97.4 | 97.4 | 97.4 | 39.4 | 11.3 | 11.3 | 11.3 | 11.3 |
| L-M | 1000 | $\alpha = 5.0$ | 10 | 10 | 93.9 | 88.3 | 96.2 | 97.8 | 97.8 | 69.3 | 98.0 | 11.3 | 11.3 | 11.3 | 11.3 | 11.3 |
| L-M | 1000 | $\alpha = 0.5$ | 10 | 10 | 92.6 | 87.9 | 94.9 | 97.9 | 97.3 | 11.3 | 97.9 | 11.3 | 11.3 | 11.3 | 11.3 | 11.3 |
| L-M | 1000 | $\alpha = 0.2$ | 10 | 10 | 82.9 | 96.7 | 93.4 | 96.8 | 96.5 | 95.6 | 96.8 | 11.4 | 11.3 | 11.3 | 11.3 | 11.3 |
| L-M | 1000 | $C = 5$ | 10 | 10 | 91.9 | 88.3 | 95.0 | 98.0 | 97.7 | 77.7 | 98.0 | 11.3 | 9.8 | 11.3 | 11.3 | 11.3 |
| L-M | 1000 | $C = 2$ | 10 | 10 | 70.8 | 96.9 | 88.4 | 97.1 | 96.8 | 11.3 | 87.6 | 11.3 | 11.3 | 11.3 | 11.3 | 11.3 |

Table 8: The detailed results of FL and SL with different learning rates on Fashion-MNIST dataset. We average the test accuracy over the last 100 rounds from 1000 total rounds when $E = 1$; average the test accuracy over the last 10 rounds from 100 total rounds when $E = 10$.

| Tag | $N$ | Dist. | $E$ | $b$ | 0.0005 FL | 0.0005 SL | 0.001 FL | 0.001 SL | 0.005 FL | 0.005 SL | 0.01 FL | 0.01 SL | 0.05 FL | 0.05 SL | 0.1 FL | 0.1 SL |
|-----|-----|-------|-----|-----|------|------|------|------|------|------|------|------|------|------|------|------|
| L-F | 1000 | IID | 1 | 10 | 54.4 | 75.0 | 68.8 | 85.9 | 83.7 | 88.4 | 86.1 | 88.2 | 88.1 | 82.4 | 87.8 | 10.0 |
| L-F | 1000 | $\alpha = 10.0$ | 1 | 10 | 57.4 | 84.0 | 71.0 | 86.5 | 84.4 | 88.8 | 86.4 | 88.6 | 88.1 | 80.2 | 87.5 | 10.2 |
| L-F | 1000 | $\alpha = 5.0$ | 1 | 10 | 57.4 | 84.6 | 74.0 | 86.3 | 84.2 | 88.6 | 86.3 | 88.4 | 88.2 | 79.9 | 87.6 | 10.0 |
| L-F | 1000 | $\alpha = 0.5$ | 1 | 10 | 59.3 | 83.4 | 72.9 | 85.4 | 83.0 | 87.6 | 85.2 | 87.3 | 86.7 | 15.6 | 85.2 | 10.0 |
| L-F | 1000 | $\alpha = 0.2$ | 1 | 10 | 60.8 | 81.6 | 71.3 | 84.3 | 80.8 | 85.9 | 83.5 | 84.9 | 83.4 | 10.0 | 10.0 | 10.0 |
| L-F | 1000 | $C = 5$ | 1 | 10 | 53.8 | 82.4 | 67.0 | 85.6 | 81.7 | 88.0 | 84.6 | 88.1 | 87.5 | 10.0 | 86.8 | 10.0 |
| L-F | 1000 | $C = 2$ | 1 | 10 | 50.9 | 80.4 | 62.6 | 83.9 | 77.5 | 87.3 | 76.5 | 86.6 | 81.1 | 10.0 | 83.0 | 10.0 |
| L-F | 1000 | IID | 10 | 10 | 75.6 | 82.7 | 78.8 | 84.0 | 84.0 | 82.5 | 85.0 | 80.0 | 46.4 | 11.0 | 10.0 | 10.0 |
| L-F | 1000 | $\alpha = 10.0$ | 10 | 10 | 76.0 | 83.6 | 79.4 | 84.1 | 84.0 | 81.7 | 84.8 | 63.7 | 10.0 | 10.0 | 10.0 | 10.0 |
| L-F | 1000 | $\alpha = 5.0$ | 10 | 10 | 76.0 | 83.2 | 79.5 | 76.6 | 83.9 | 80.8 | 84.5 | 77.0 | 10.0 | 10.0 | 10.0 | 10.0 |
| L-F | 1000 | $\alpha = 0.5$ | 10 | 10 | 72.7 | 78.0 | 75.8 | 79.2 | 82.8 | 76.8 | 83.8 | 10.0 | 10.0 | 10.0 | 10.0 | 10.0 |
| L-F | 1000 | $\alpha = 0.2$ | 10 | 10 | 65.1 | 79.0 | 68.4 | 78.6 | 79.3 | 70.9 | 80.4 | 10.0 | 10.0 | 10.0 | 10.0 | 10.0 |
| L-F | 1000 | $C = 5$ | 10 | 10 | 68.6 | 78.9 | 72.7 | 80.1 | 80.6 | 77.5 | 82.7 | 25.8 | 78.0 | 10.0 | 10.0 | 10.0 |
| L-F | 1000 | $C = 2$ | 10 | 10 | 52.8 | 72.0 | 59.5 | 72.3 | 73.3 | 10.0 | 75.9 | 10.0 | 34.5 | 10.0 | 10.0 | 10.0 |

Table 9: The detailed results of FL and SL with different learning rates on CIFAR-10 dataset. We average the test accuracy over the last 400 rounds from 4000 total rounds when $E = 1$; average the test accuracy over the last 40 rounds from 400 total rounds when $E = 10$. We don not execute the experiments whose learning rates are larger than the "threshold" learning rate ("-" in the table).

| Tag | $N$ | Dist. | $E$ | $b$ | 0.0005 | | 0.001 | | 0.005 | | 0.01 | | 0.05 | | 0.1 | |
|---|---|---|---|---|---|---|---|---|---|---|---|---|---|---|---|---|
| | | | | | FL | SL | FL | SL | FL | SL | FL | SL | FL | SL | FL | SL |
| V-10 | 500 | IID | 1 | 10 | - | - | 43.1 | 86.2 | 84.0 | 87.0 | 85.5 | 85.1 | 86.4 | 10.0 | 10.0 | 10.0 |
| V-10 | 500 | $\alpha = 10.0$ | 1 | 10 | - | - | 47.4 | 86.4 | 84.2 | 86.9 | 85.7 | 84.2 | 86.3 | 10.0 | 10.0 | - |
| V-10 | 500 | $\alpha = 5.0$ | 1 | 10 | - | - | 47.2 | 86.5 | 84.2 | 87.0 | 85.6 | 84.5 | 86.1 | 10.0 | 10.0 | - |
| V-10 | 500 | $\alpha = 0.5$ | 1 | 10 | - | - | 44.5 | 85.4 | 82.1 | 85.5 | 84.1 | 10.0 | 10.0 | - | - | - |
| V-10 | 500 | $\alpha = 0.2$ | 1 | 10 | 22.9 | 81.2 | 39.4 | 83.5 | 78.1 | 83.0 | 80.5 | 10.0 | 10.0 | - | - | - |
| V-10 | 500 | $C = 5$ | 1 | 10 | 14.4 | 83.7 | 37.6 | 85.9 | 82.3 | 86.5 | 84.7 | 83.4 | 85.5 | - | 10.0 | 10.0 |
| V-10 | 500 | $C = 2$ | 1 | 10 | 14.0 | 81.9 | 23.9 | 84.7 | 73.7 | 84.7 | 79.2 | 12.9 | 80.0 | 10.0 | 10.0 | - |
| V-10 | 500 | IID | 10 | 10 | 51.5 | 83.2 | 68.3 | 83.7 | 77.3 | 78.3 | 77.7 | 10.0 | 10.0 | 10.0 | 10.0 | 10.0 |
| V-10 | 500 | $\alpha = 10.0$ | 10 | 10 | 53.3 | 82.8 | 69.1 | 83.0 | 76.3 | 10.0 | 77.9 | 10.0 | 10.0 | - | 10.0 | - |
| V-10 | 500 | $\alpha = 5.0$ | 10 | 10 | 54.2 | 82.1 | 68.1 | 82.7 | 77.6 | 10.0 | 76.9 | 10.0 | 10.0 | - | - | - |
| V-10 | 500 | $\alpha = 0.5$ | 10 | 10 | 40.6 | 75.1 | 58.4 | 76.9 | 71.0 | 10.0 | 71.8 | 10.0 | 10.0 | 10.0 | 10.0 | 10.0 |
| V-10 | 500 | $\alpha = 0.2$ | 10 | 10 | 27.1 | 65.0 | 41.0 | 10.0 | 66.2 | 10.0 | 66.9 | 10.0 | 10.0 | 10.0 | 10.0 | 10.0 |
| V-10 | 500 | $C = 5$ | 10 | 10 | 36.3 | 77.6 | 59.6 | 78.9 | 73.9 | 10.0 | 74.2 | - | 10.0 | - | 10.0 | - |
| V-10 | 500 | $C = 2$ | 10 | 10 | 20.3 | 58.7 | 27.8 | 10.0 | 51.8 | 10.0 | 61.6 | - | 10.0 | - | 10.0 | - |

