# OpenReview forum: "Convergence Analysis of Split Learning on Non-IID Data"
_ICLR.cc/2023/Conference — Submitted to ICLR 2023_

### Official Review · Reviewer_4bP6 · 2022-10-21

**Confidence:** 4
**Correctness:** 3
**Technical Novelty And Significance:** 2
**Empirical Novelty And Significance:** 2
**Recommendation:** 5

**Clarity, Quality, Novelty And Reproducibility:**

The clarity can be improved, for example,
- The notations in Eq. 2 are not well defined until Algorithm 1, it is better to reorder the contents to ensure a clearer flow of reading.
- In section 4.2, the paragraph "Client drift in SL is upper bounded, as shown...." does not have a clear flow of ideas.
- $\eta_{SL}$ and $\eta_{FL}$ are not clearly defined.

**Strength And Weaknesses:**

Strengths:
- The analysis seems solid and correct, though I didn't go through all the details in the appendix.
- To my knowledge, the convergence result is new, which complements the theoretical study of split learning.

Weaknesses:
- The main takeaway of this work is that SL is faster than FL and generalizes similar to FL on mildly non-IID data, while FL prevails on highly non-IID data. This conclusion does not seem surprising to me since previous works (e.g., Gao et al. (2020; 2021) cited in the submission) already made such observations.
- For the theoretical results, since split learning runs sequentially for its local updates, it is basically minibatch SGD with biased updates. Thus, as long as the accumulated error (client drift) can be bounded, establishing its convergence seems straightforward to me. I hope the authors can discuss the technical difficulties on deriving the convergence bound.
- The discussion (or guidance for hyperparameter selection) for the theorem is inadequate. For example, in section 4.1, "one needs to meticulously design the value of K for the best performance", "large value of K forces SL prone to converge to local optima", "to reduce the communication overhead without hurting the convergence rate" all require further discussion, such as to analyze the communicated bits of split learning. The Effect of $\eta_g$ seems interesting, but it is not thoroughly studied. It seems that we can choose an arbitrarily large $\eta_g$?
- The presentation also needs to be improved (see below).


**Summary Of The Paper:**

This work establishes the convergence bound of split learning (SL) for non-convex objectives with non-iid data. The convergence results provide some insight on the parameter tuning and potential benefits/limitations of split learning, compared with standard federated learning (FL, local SGD). The authors also conduct empirical comparison between spilt learning and federated learning with respect to data heterogeneity, which provides a deeper understanding between them.

**Summary Of The Review:**

This work provides some new analysis of SL on non-iid data and compare it with FL, which is only marginally interesting from my perspective. It is not clear what are the technical difficulties of proving the theorems, and the discussion of the theorems is inadequate. The overall presentation also needs to be improved.

---------------------------------------------------
After rebuttal: Thank the authors for the detailed responses and revision, I really appreciate the effort. I have also checked other reviews and responses. The revision greatly improved the clarity of this paper. However, I still think the novelty of this paper is below the bar as there seems to be no strong or really exciting result. I keep my score but will not be upset if this work gets accepted.

---

> ### Author Response · Authors · 2022-11-22
> **Response to Reviewer 4bP6 (Part 3/3)**
>
> ##### **Q3. The discussion for the theorem is inadequate.**
>
> > The discussion (or guidance for hyperparameter selection) for the theorem is inadequate. For example, in section 4.1, "one needs to meticulously design the value of K for the best performance", "large value of K forces SL prone to converge to local optima", "to reduce the communication overhead without hurting the convergence rate" all require further discussion, such as to analyze the communicated bits of split learning. The Effect of $\eta_g$ seems interesting, but it is not thoroughly studied. It seems that we can choose an arbitrarily large $\eta_g$?
>
> Thanks for checking our work carefully. In the first version, we make some less rigorous presentations. Some immature sentences are removed in the revised version. Next, we explain some points in the revised paper.
>
> + **For $K$**. In the discussion of Section 4.2, Theorem 1 shows that (i) the optimal $K$ exists (Over-large or Over-small are not suggested). This has been evaluated in Fig. 2 (b). (ii) larger data heterogeneity makes the optimal $K$ smaller. This has been also evaluated in Table 2. Furthermore, $K$ makes a big difference to the comparison between FL and SL (see the 1-st point of the responses). So we think that the discussion about $K$ in the revised version is adequate.
> + **For $\eta_g$**. We have provided one figure at Fig. 2 (c) and more discussion in Section 5.2 and figures in Appendix E. We observe that tuning $\eta_g$ in the limited range seems effective, especially when $\eta_l$ is small. But $\eta_g$ can not be set infinitely too large in practice. Unfortunately, there is still a big gap between theory and practice both in FL [7] and SL, which we think is beyond this work and deserves further research.
>
> ##### **Q4. The presentation also needs to be improved.**
>
> Thank you for the suggestions on improving the clarity of our paper. We have improved our writing based on your suggestions in the revised version.
>
> + **The notations in Eq. 2 are not well defined until Algorithm 1**. As suggested, we have rewritten the contents of this portion. Besides, we have provided one table of notations in Appendix A (Table 3).
> + **In section 4.2, the paragraph "Client drift in SL is upper bounded, as shown...." does not have a clear flow of ideas**. As suggested, we have removed some confusing sentences in Section 4.2 in the revised version.
> + **$\eta_{FL}$ and $\eta_{SL}$ are not clearly defined**. We introduce the effective learning rates ($\tilde{\eta}\_{\text{FL}} = K\eta_g\eta_l$ ,$\tilde{\eta}\_{\text{SL}} = NK\eta_g\eta_l$) based on [4, 5] in Section 4.3 and define these variables explicitly in the revised version.
>
> ---
>
> We hope that our response addresses your concerns. We are glad to answer any further questions. Thank you!
>
> ### References
>
> [1] Kairouz, P., McMahan, H. B., Avent, B., Bellet, A., Bennis, M., Bhagoji, A. N., ... & Zhao, S. (2021). Advances and open problems in federated learning. *Foundations and Trends® in Machine Learning*, *14*(1–2), 1-210.
>
> [2] Ajalloeian, A., & Stich, S. U. (2020). On the convergence of SGD with biased gradients. *arXiv preprint arXiv:2008.00051*.
>
> [3] Khaled, A., Mishchenko, K., & Richtárik, P. (2020, June). Tighter theory for local SGD on identical and heterogeneous data. In *International Conference on Artificial Intelligence and Statistics* (pp. 4519-4529). PMLR.
>
> [4] Karimireddy, S. P., Kale, S., Mohri, M., Reddi, S., Stich, S., & Suresh, A. T. (2020, November). Scaffold: Stochastic controlled averaging for federated learning. In *International Conference on Machine Learning* (pp. 5132-5143). PMLR.
>
> [5] Wang, J., Liu, Q., Liang, H., Joshi, G., & Poor, H. V. (2020). Tackling the objective inconsistency problem in heterogeneous federated optimization. *Advances in neural information processing systems*, *33*, 7611-7623.
>
> [6] Lian, X., Zhang, C., Zhang, H., Hsieh, C. J., Zhang, W., & Liu, J. (2017). Can decentralized algorithms outperform centralized algorithms? a case study for decentralized parallel stochastic gradient descent. *Advances in Neural Information Processing Systems*, *30*.
>
> [7] Anonymous. Fedexp: Speeding up federated averaging via extrapolation. In Submitted to The Eleventh International Conference on Learning Representations, 2023. URL https://openreview.net/forum?id=IPrzNbddXV. under review

---

> ### Author Response · Authors · 2022-11-22
> **Response to Reviewer 4bP6 (Part 2/3)**
>
> ##### **Q2. Difficulties**
>
> > For the theoretical results, since split learning runs sequentially for its local updates, it is basically minibatch SGD with biased updates. Thus, as long as the accumulated error (client drift) can be bounded, establishing its convergence seems straightforward to me. I hope the authors can discuss the technical difficulties on deriving the convergence bound.
>
> Thank you for giving another method for the convergence of SL. However, biased SGD only converges to a neighborhood around the stationary point of the global function according to [2]. Furthermore, through this way, Theorem 3 (Bounding the progress of one client) is given in Appendix C, D for comparison, which shows that Theorem 3 only converges to a neighborhood of which the size is determined by the heterogeneity. (Table 6 summarizes all the theories of this paper in Appendix C)
>
> We derive the convergence guarantee based on the theory in the FL literature [3, 4, 5], since these two algorithms have many similarities. The main difficulties are shown below:
>
> + **Bounding the progress of all clients in one round is necessary.** As shown in our theory, we bound the progress of all clients, $\mathbb{E} [f(\mathbf{x}^{r+1}) - f(\mathbf{x}^{r})]$ (Theorem 1), instead of bounding the progress of one client, $\mathbb{E} [f(\mathbf{x}_{i+1}^{(r,0)}) - f(\mathbf{x}_i^{(r,0)})]$ (Theorem 3). This is the most critical point of our proof.
>
>   Bounding the progress of one client (Theorem 3). It is immediate to bound $\mathbb{E} [f(\mathbf{x}\_{i+1}^{(r,0)}) - f(\mathbf{x}\_i^{(r,0)})]$ with the Assumption $\mathbb{E}\_{i \sim \mathcal{U}([N])} \left[\\|\nabla f_i (\mathbf{x}) - \nabla f(\mathbf{x})\\|^2\right] \leq G^2
>   $ , where $i$ is uniformly sampled from $\{1, \ldots, N\}$. But it only converges to a neighborhood, as discussed above.
>
>   However, as proved by Theorem 1, SL can converge to the stationary point by bounding the progress of all clients in one round.
>
> + **Bounding the global variance** $\mathbb{E}\left[\\|\sum\_{i=1}^N\sum\_{k=0}^{K-1}\left[\mathbf{g}\_i(\mathbf{x}\_i^{(r,k)})-\nabla f\_i(\mathbf{x}\_i^{(r,k)})\right]\\|^2\right]$. In SL, models are trained in sequence and the data is non-IID, so Jensen's inequality is used to get:
>   $$
>   \mathbb{E}\left[\\|\sum\_{i=1}^N\sum\_{k=0}^{K-1}\left[\mathbf{g}\_i(\mathbf{x}\_i^{(r,k)})-\nabla f\_i(\mathbf{x}\_i^{(r,k)})\right]\\|^2\right] \leq N \sum\_{i=1}^N \mathbb{E}\left[\\|\sum\_{k=0}^{K-1}\left[\mathbf{g}\_i(\mathbf{x}\_i^{(r,k)})-\nabla f\_i(\mathbf{x}\_i^{(r,k)})\right]\\|^2\right]
>   $$
>   Then we use one interesting way to bound the term.
>   $$
>   N \sum\_{i=1}^N \mathbb{E}\left[\\|\sum\_{k=0}^{K-1}\left[\mathbf{g}\_i(\mathbf{x}\_i^{(r,k)})-\nabla f\_i(\mathbf{x}\_i^{(r,k)})\right]\\|^2\right] = N \sum\_{i=1}^N \mathbb{E}\left[\\|\sum\_{k=0}^{K-1}\left[\mathbf{g}\_i(\mathbf{x}\_i^{(r,k)})-\nabla f\_i(\mathbf{x}\_i^{(r,k)})\right]\\|^2 \vert \mathbf{x}\_i^{(r,0)} \right]\leq N^2K \sigma^2,
>   $$
>
>   where we use the law of total expectation in the equality, Lemma 3 [4, 5] (Appendix D) and Assumption 2 in the inequality. As shown that it is tighter than using Jensen's inequality directly ($N^2K^2\sigma^2$).
>
> + **Bounding the client drift** $\sum\_{i=1}^N \sum\_{k=0}^{K-1} \mathbb{E}\left[\\|\mathbf{x}\_i^{(r,k)} - \mathbf{x}^{r}\\|^2\right]$. Note that although we use the method developed in [3, 4, 5], it is more difficult to bound the client drift than FL. Since client $i$ initialize with $\mathbf{x}_{i-1}^{(r,0)}$ instead of $\mathbf{x}^{r}$.
>
> We hope the response can address your concerns about the difficulties of our work.

---

> ### Author Response · Authors · 2022-11-22
> **Response to Reviewer 4bP6 (Part 1/3)**
>
> Thanks for your careful checking our paper. We greatly appreciate your positive comments on the solid theory of our work. We have revised our paper based on your insightful suggestions and highlighted the major changes in blue.
>
> ##### **Q1. The main takeaway**
>
> > The main takeaway of this work is that SL is faster than FL and generalizes similar to FL on mildly non-IID data, while FL prevails on highly non-IID data. This conclusion does not seem surprising to me since previous works (e.g., Gao et al. (2020; 2021) cited in the submission) already made such observations.
>
> We have improved our theory and experiments in the revised version. The key contributions are listed below to stand this work out from the previous work (We mark the new contribution in our revised version).
>
> **Theoretically.**
>
> + We give the convergence guarantee of SL, which shows a convergence rate of $\mathcal{O}(1/\sqrt{T})$.
> + Based on the convergence guarantee, we analyze the effect of some factors (e.g., data heterogeneity and local update steps).
> + By the convergence guarantee, we show that the constraint on the local learning rate of SL is tougher than FL, which is significant for the learning rate selection of SL.
> + **The guarantee of SL is worse than FL in terms of training rounds on non-IID data.**
>
> **Empirically.** For comparison with FL, We adopt the cross-device settings [1] in the revised version, where clients are IoT devices and the client number can be very large even to $10^{10}$ (this is the environment that SL actually works in).
>
> + Experiments are provided to verify the convergence theory of SL (the 2-nd point of theory contributions).
> + The best and threshold learning rate of SL is smaller than FL (the 3-rd point of theory contributions). Note that in Gao et al. (2020; 2021), they use the same learning rate for FL and SL in their experiments, so their conclusions may be limited.
> + **The performance of SL is worse than FL when the number of local update steps is large on highly non-IID data in cross-device settings**  (the 4-th point of theory contributions).
> + **The performance of SL can be better than FL when choosing the small number of local update steps on highly non-IID data in cross-device settings.** (it implies that SL would be the preferable choice for the computation-limited scenario, where the computation resources of the local devices become the bottleneck of the training system.)
>
> At last, our work can bridge the gap between FL and SL and provide deep understanding of these two approaches.
>
> We hope the new generalized conclusions can address your concerns about the novelty of our work.

---

### Official Review · Reviewer_434u · 2022-10-22

**Confidence:** 4
**Correctness:** 4
**Technical Novelty And Significance:** 4
**Empirical Novelty And Significance:** 4
**Recommendation:** 6

**Clarity, Quality, Novelty And Reproducibility:**

I think the work is novel and important. I like the presentation and clarity of the paper. In Sec.3, the paper gives one sentence summary of techniques of each paper. Later, following each theorem, the paper presents plenty of discussions.
I suggest adding a table of notations, and a table of main techniques (the iteration expression) and rates of the prior works in appendix. The SplitFedv2/3/SFLG can be explained more clearly.

**Strength And Weaknesses:**

The topic is of great importance for theoretical analysis of optimization in distributed machine learning. I believe the proof is correct and solid. The experiments are concrete.

Some questions:
- I suggest a table for notations, and highlight the super/sub-scripts of $x$.
- Eq (2) is correct but a bit misleading for readers. Although there is a sum over $k$, one cannot get all $x_i^{r,k}$ over $k$ at the same time because the $k$ stands for iterations, so the $x_i^{r,k}$ comes sequentially like in Algo 1 box. When I saw a sum, I felt the terms can somehow be parallelized but actually not, so it would be great to clarify in a footnote, or always write as in Algo 1.
- Assumption 3: Could you explain why “In the IID case, $B = 1, G = 0$”? I suspect (e.g. in stochastic case) $E\|\nabla f_i\|^2 = \|\nabla f\|^2 + Cov(\nabla f)$, was it wrong? Could you use a few simple examples in appendix calculating $B,G$, like linear functions, $f(x) = ax$ and $f_i(x) = a_i(x)$, and so on?
- Step size: you choose $T=NKR$, where is $B$? In Thm.1, the third term in $\min$ depends on $B$.
- Rate: Is the rate outer loop complexity, and the total gradient calculation $K$ times that number? Could you compare it with SGD, and GD if you simply calculate all stochastic gradients together and average them? I didn’t mean SL has to beat them, but it would be helpful and interesting to compare. If the rate is not as good, it would be great to explain the reason. Is it due to the fact that you cannot shuffle or randomize the individual/local gradients? Are you aware of any lower bound of RHS of Cor. 1 and how suboptimal the result is from that optimal rate?
- Constant step size: if the step size depends on $T$ and error depends on $T$, would it be better to choose a larger error and a small $T$ at the beginning, i.e. use a large step size at the beginning, then refine the error, increase $T$ and decrease the step size? In SGD one can use the decreasing step size.


**Summary Of The Paper:**

This paper analyzes the convergence rate of split learning theoretically. With constant step sizes in both inner and outer loops, the outer loop complexity is $T^{-1/2}$ with additional terms depending on noise and dissimilarity. Then it compares the rate between SL and FL and presents discussions/explanations.

**Summary Of The Review:**

I think the paper is important and solid, and the clarity is great. With a few more revision regarding my questions above, I believe it's ready for acceptance.

---

> ### Author Response · Authors · 2022-11-22
> **Response to Reviewer 434u (Part 3/3)**
>
> ##### **Q6. Clarity and Notations**
>
> > I suggest a table for notations, and highlight the super/sub-scripts of $x$.
>
> > I think the work is novel and important. I like the presentation and clarity of the paper. In Sec.3, the paper gives one sentence summary of techniques of each paper. Later, following each theorem, the paper presents plenty of discussions. I suggest adding a table of notations, and a table of main techniques (the iteration expression) and rates of the prior works in appendix. The SplitFedv2/3/SFLG can be explained more clearly.
>
> Thanks for your appreciation and advice for clarity of our work. As suggested, we have provided more details in Appendix in the revised version, including:
>
> + More details (including communication cost of SL) about SL and a table for notations in Appendix A.
> + The update rules (the iteration expression) of Minibatch SGD, FL and SL are summarized in Table 5 in Appendix B.
> + SplitFedv2/3/SFLG/FedSeq are explained in detail in Appendix B.
> + All the theories of this paper is also summarized in Table 6 in Appendix C.
>
> ---
>
> We are glad to answer any further questions. Thanks for your appreciation for our work!
>
> ### References
>
> [1] Karimireddy, S. P., Kale, S., Mohri, M., Reddi, S., Stich, S., & Suresh, A. T. (2020, November). Scaffold: Stochastic controlled averaging for federated learning. In *International Conference on Machine Learning* (pp. 5132-5143). PMLR.
>
> [2] Bottou, L., Curtis, F. E., & Nocedal, J. (2018). Optimization methods for large-scale machine learning. *Siam Review*, *60*(2), 223-311.

---

> > ### Comment · Reviewer_434u · 2022-11-23
> > **Thanks for the comment**
> >
> > I do not have more questions regarding this part.

---

> ### Author Response · Authors · 2022-11-22
> **Response to Reviewer 434u (Part 2/3)**
>
> ##### **Q3. Step size in Cor. 1**
>
> > Step size: you choose $T=NKR$, where is $B$? In Thm.1, the third term in $\min$ depends on $B$.
>
> Thanks for pointing this out. Actually, $T:=NKR$ is the definition of $T$, where $T$ is the total iterations, $N$ is the number of clients, $K$ is the number of local update steps and $R$ is the total training rounds. In our revised version, we give the detailed requirement of $T \geq 4N^2K^2 \max\\{\frac{2B^2+1}{\eta_g^2}, 1\\}$ for $T$ when choosing $\eta_g\eta_l = \frac{1}{L\sqrt{T}}$ (Corollary 1 in the revised version).
>
> ##### **Q4. Rate**
>
> > Rate: Is the rate outer loop complexity, and the total gradient calculation $K$ times that number? Could you compare it with SGD, and GD if you simply calculate all stochastic gradients together and average them? I didn’t mean SL has to beat them, but it would be helpful and interesting to compare. If the rate is not as good, it would be great to explain the reason. Is it due to the fact that you cannot shuffle or randomize the individual/local gradients? Are you aware of any lower bound of RHS of Cor. 1 and how suboptimal the result is from that optimal rate?
>
> + **Convergence rate**. The rate in Corollary 1 is the rate in terms of the total iterations $T$ ($\frac{1}{\sqrt{T}}$, or $\frac{1}{\sqrt{NKR}}$). We also give the rate in terms of the total training rounds $R$ (i.e., the outer loop complexity, $\frac{1}{\sqrt{R}}$) in Corollary 2 (Table 6 summarizes all the theories of this paper in Appendix C in the revised version).
> + **Compare with SGD and GD**. The convergence guarantee of SL is worse than SGD. Compared with the SGD, the $T_2$ client drift error term (In Theorem 1 of the revised version) is the additional term caused by the client-drift in SL. But we can recover the convergence of SGD [2] with $N=1$ and $K=1$.
> + **The reason for the sub-optimal result**. Yes. "You cannot shuffle or randomize the individual/local gradients" is actually the reason, which will cause the "client drift" problem in FL and SL [1].
> + **Lower bound**. Thank you for pointing this out. The lower bound is very useful to check the tightness of convergence result. We promise to strengthen our convergence analysis to incorporate the lower bound of SL in the future.
>
> ##### **Q5. Constant step size**
>
> > Constant step size: if the step size depends on $T$ and error depends on $T$, would it be better to choose a larger error and a small $T$ at the beginning, i.e. use a large step size at the beginning, then refine the error, increase $T$ and decrease the step size? In SGD one can use the decreasing step size.
>
> Yes. Decreasing step size is adopted in SGD and used widely in the practical ML task, which is also currently a theoretical limitation of our convergence bound. At the same time, we find that most work in the FL literature use the fixed step size for the proof of the non-convex function, so this limitation does not affect the comparison between FL and SL. We also give one more tighter bound in Corollary 3 by choosing the fixed step size carefully (Table 6 summarizes all the theories of this paper in Appendix C) based on the technique in [1].
>
> Choosing the $NK\eta_g\eta_l=\frac{1}{\sqrt{R}}$ (Corollary 2):
>
> $$
> \mathbb{E}[\|\nabla f(\bar{\mathbf{x}}^R)\|^2] \leq \mathcal{O}\left(\frac{F}{\sqrt{R}}\right)+\mathcal{O}\left( \frac{KG^2 + \sigma^2}{KR} \right) + \mathcal{O}\left(\frac{\sigma^2}{K\sqrt{R}}\right)
> $$
>
> Choosing the fixed step size carefully (Corollary 3):
>
> $$
> \mathbb{E}[\|\nabla f(\bar{\mathbf{x}}^R)\|^2] \leq \mathcal{O}\left(\frac{F\sqrt{(2B^2+1)}}{R+1}\right)+\mathcal{O}\left( \frac{F^{\frac{2}{3}}(G^2+\frac{\sigma^2}{K})^{\frac{1}{3}}}{(R+1)^{\frac{2}{3}}} \right) + \mathcal{O}\left(\frac{F\sigma^2}{\sqrt{K(R+1)}}\right),
> $$
>
> where $F \coloneqq f(\mathbf{x}^0) - f(\mathbf{x}^*)$, $R$ is the total rounds, $\bar{\mathbf{x}}^R= \frac{1}{R}\sum_{r=0}^{R-1}\mathbf{x}^r$ is the averaged global model over the $R$ rounds, $N$ is the number of clients, $K$ is the local update steps. Note that these two bounds are all in terms of total training rounds (i.e., the outer loop complexity).

---

> > ### Comment · Reviewer_434u · 2022-11-23
> > **Constant step size**
> >
> > I would guess that using a decreasing step size would likely give a better rate, and we probably prefer a less-sub-optimal rate if there is not a big technical difficulty.
> >
> > I suggest starting with an epoch based method: Set error target as $E_0$, and run a few iterations, and set error target as $E_1:= E_0/2$, run another few iterations, until ending up with the final error $E$. In total, you have $\log_2(E_0/E)$ outer epochs, and you can sum up the number of iterations in each inner loop. I guess this is not difficult and perhaps the rate (total iterations) is better, and perhaps the error $E$ can even be as small as possible without the constant drifting error or global variance terms, etc. You can also resemble the idea for dividing/multiplying $2$ for other relevant parameters like $T$, $\eta$, etc.
> >
> > I have scored it as 6, already above border, so I decide not to change. In my opinion a comprehensive investigation with step sizes can make it 8.

---

> > > ### Author Response · Authors · 2022-11-23
> > > **Further response to Reviewer 434u**
> > >
> > > ##### **Q1. "deterministic/stochastic" or "IID/non-IID"**
> > >
> > > > I don't have major questions for this part. One comment: when you refer to $B=1$,$G=0$ as IID and differentiate with other cases, is it typically called "deterministic/stochastic" or "IID/non-IID"? Perhaps the convention is the latter pair but I use the former more often, if you keep the terminology, I suggest defining the terminology at least in the footnote.
> > >
> > > As suggested, we will define the terminology (IID/non-IID) explicitly in the later submission.
> > >
> > > ##### **Q2. an epoch based method**
> > >
> > > > I suggest starting with an epoch based method: Set error target as $E_0$, and run a few iterations, and set error target as $E_1:=E_0/2$, run another few iterations, until ending up with the final error $E$. In total, you have $\log_2⁡(E_0/E)$ outer epochs, and you can sum up the number of iterations in each inner loop. I guess this is not difficult and perhaps the rate (total iterations) is better, and perhaps the error $E$ can even be as small as possible without the constant drifting error or global variance terms, etc. You can also resemble the idea for dividing/multiplying $2$ for other relevant parameters like $T$, $\eta$, etc.
> > >
> > > Thanks very much for your detailed and insightful suggestions for improving this work. This epoch based method is similar to the Epoch-SGD [1], which is very exciting for SL. We will do further investigations into the epoch based method in the SGD and FL (Local SGD) literature and try to improve our convergence guarantee following this method.
> > >
> > > Thanks so much for your positive feedback on our work and response and the constructive suggestions.
> > >
> > > [1] Hazan, E., & Kale, S. (2014). Beyond the regret minimization barrier: optimal algorithms for stochastic strongly-convex optimization. *The Journal of Machine Learning Research*, *15*(1), 2489-2512.

---

> ### Author Response · Authors · 2022-11-22
> **Response to Reviewer 434u (Part 1/3)**
>
> Thanks for your careful checking our paper. We greatly appreciate your positive comments on the clarity of our writing, the rigorous theory and novelty of our work. We have revised our paper based on your insightful suggestions and highlighted the major changes in blue.
>
> ##### **Q1. Misleading Eq. (2)**
>
> > Eq (2) is correct but a bit misleading for readers. Although there is a sum over $k$, one cannot get all $x_i^{r,k}$ over $k$ at the same time because the $k$ stands for iterations, so the $x_i^{r,k}$ comes sequentially like in Algo 1 box. When I saw a sum, I felt the terms can somehow be parallelized but actually not, so it would be great to clarify in a footnote, or always write as in Algo 1.
>
> We have polished the description of Algo. 1. For Eq. (2) in the first version (Eq. (4) in the revised version) we keep the statement as Algo. 1 as suggested.
>
> ##### **Q2. Assumption 3**
>
> > Assumption 3: Could you explain why "In the IID case, $B=1$, $G=0$"? I suspect (e.g. in stochastic case) $\mathbb{E} \|\nabla f_i\|^2 = \|\nabla f\|^2 + \textrm{Cov}(\nabla f_i)$, was it wrong? Could you use a few simple examples in appendix calculating $B$, $G$, line linear functions, $f(x) = ax$ and $f_i (x) = a_i (x)$, and so on?
>
> Assumption 3 is used in the FL literature [1]. We give some explanations in the following.
>
> + **why $B=1$, $G=0$ in the IID case?** In Assumption 3, we use
>   $$
>   \frac{1}{N}\sum_{i=1}^N \lVert \nabla f_i(x) \rVert^2 = B^2 \lVert \nabla f(x) \rVert^2 + G^2
>   $$
>   to measure the heterogeneity of data distribution. In the IID case, $f_i (x) = f_j (x) = f(x), \forall i, j$. Then we can get:
>
>   $$
>   \frac{1}{N}\sum_{i=1}^N \lVert \nabla f_i(x) \rVert^2 = \lVert \nabla f(x) \rVert^2
>   $$
>   So $B=1$ and $G=0$.
>
>
> + **$\mathbb{E} \|\nabla f_i\|^2 = \|\nabla f\|^2 + \textrm{Cov}(\nabla f_i)$**. Considering one extreme case of Assumption 3 where $B=1$, then we have:
>
>   $$
>   \frac{1}{N}\sum_{i=1}^N \lVert \nabla f_i(x) \rVert^2 - \lVert \nabla f(x) \rVert^2 \leq G^2\\
>   \Rightarrow \frac{1}{N}\sum_{i=1}^N \lVert \nabla f_i(x) - f(x) \rVert^2 \leq G^2
>   $$
>
>   where we use $\nabla f(x) = \frac{1}{N}\sum_{i=1}^N f_i(x)$ and $\mathbb{E} [\\|x - \mathbb{E}[x]\\|^2] = \mathbb{E}[x^2] - \\|\mathbb{E}[x]\\|^2 $. In other words, the extreme case of Assumption 3 bound the variance of local objective function. This is also $\mathbb{E} \|\nabla f_i\|^2 = \|\nabla f\|^2 + \textrm{Cov}(\nabla f_i)$ you have mentioned. In fact, Assumption 3 is weaker than this extreme case.
>
>
> + **A few of simple example**:
>
>   | $f_1 (x)$            | $f_2 (x)$            | $f(x)$            | $B^2$ | $G^2$ |
>   | -------------------- | -------------------- | ----------------- | ----- | ----- |
>   | $x$                  | $-x$                 | 0                 | 1     | 1     |
>   | $\frac{1}{2}(x-1)^2$ | $\frac{1}{2}(x+1)^2$ | $\frac{x^2+1}{2}$ | 1     | 1     |
>   | $\frac{1}{2}(x-1)^2$ | $\frac{3}{2}(x+1)^2$ | $x^2+x+1$         | 2     | 3     |

---

> > ### Comment · Reviewer_434u · 2022-11-23
> > **Thanks for the comment.**
> >
> > I don't have major questions for this part. One comment: when you refer to $B=1,G=0$ as IID and differentiate with other cases, is it typically called "deterministic/stochastic" or "IID/non-IID"? Perhaps the convention is the latter pair but I use the former more often, if you keep the terminology, I suggest defining the terminology at least in the footnote.

---

### Official Review · Reviewer_SLb6 · 2022-10-24

**Confidence:** 3
**Clarity, Quality, Novelty And Reproducibility:** See Below
**Correctness:** 3
**Technical Novelty And Significance:** 2
**Empirical Novelty And Significance:** 2
**Recommendation:** 6

**Strength And Weaknesses:**

See Below

**Summary Of The Paper:**

The paper derives the convergence guarantee of SL for non-convex objectives on non-IID data.  It compares also SL against FL theoretically and empirically,

**Summary Of The Review:**

-In Algo 1. I have a problem understanding exactly what is the communication round here, because in step 8 and 10 there is communication between clients and server. You don't count that as a communication round?

To have the local update in step 12 (local model update) according to algo 1, communication is needed between the server and the client, so what is local here since there is a communication between server and client?...

It seems that there are "local" communication rounds which are denoted by k=0.1,...K and there are "global" communication rounds which are r=0.1...R can you clarify what this is, because for me the communication happens for both iterations k and r? am i missing something?

-In step 12 you put, client-side and server-side updates, again I'm confused here, how the server can access the local gradients to do this local update.

- In step 3, since in SL things work in parallel, why do you need to sample a subset of clients only?

-Table 1: in local updates for SL, is it a typo of indices there x_{i}^{r,k+1} = x_{i-1}^{r,K}- ... or am I missing something again?

-if my understanding is correct, x^r of FL is the same as x^{r,K} of SL. Then from Table 1, SL and FL and producing the same iterations if they use the same learning rates and batches? if my understanding is wrong, can you tell what prevents SL from using x^r in local updates.

-The reported complexity bounds between SL and FL are the same up to the multiplicative constant N.
The reason is, I think, the choice of learning rates.
If you change the lr of SL by the lr of FL over N you get the same bounds for FL and SL!

-Does K depend on i? you have K=En_i/b.


------after rebuttal----

After discussions with authors and rebuttal with other reviewers. The authors addressed some of my concerns. I still think that the complexity results given in the first version of the paper, SL and FL have exactly the same complexity bounds by choosing the right learning rates. See my comments and discussions on this. I decided to increase my rating of the paper to 6.

---

> ### Author Response · Authors · 2022-11-22
> **Response to Reviewer SLb6 (Part 2/2)**
>
> ##### **Q4. Table 1**
>
> > Table 1: in local updates for SL, is it a typo of indices there x_{i}^{r,k+1} = x_{i-1}^{r,K}- ... or am I missing something again?
>
> > if my understanding is correct, x^r of FL is the same as x^{r,K} of SL. Then from Table 1, SL and FL and producing the same iterations if they use the same learning rates and batches? if my understanding is wrong, can you tell what prevents SL from using x^r in local updates.
>
> Thanks for careful checking Table 1, but it is not a typo. As suggested, the table has been deferred to Appendix B (Table 5) for a more explicit presentation. Clients are trained in sequence in SL. So the first client in round $r$ will initialize with $\mathbf{x}_c^{r}$. But the other clients in round $r$ will initialize with the previous client's model parameters. The server-side model is alike. So the process can be stated as (the concatenation of the client-side model and the server-side model):
>
> $$
> \mathbf{x}\_{i}^{(r,0)} \leftarrow
> \begin{cases}
> \mathbf{x^r}, i = 1\\\\
> \mathbf{x}\_{i-1}^{(r,K)}, i > 1
> \end{cases}
> $$
>
> So for SL, the concatenation of the client-side model and the server-side model actually start training from the $\mathbf{x}_{i-1}^{(r,K)}$, which is the main difference with FL.
>
> ---
>
> ##### **Q5. The bounds are the same for FL and SL**
>
> > The reported complexity bounds between SL and FL are the same up to the multiplicative constant N. The reason is, I think, the choice of learning rates. If you change the lr of SL by the lr of FL over N you get the same bounds for FL and SL!
>
> Thanks for pointing this out. Fortunately, it does not affect our convergence rate. We have improved our theory and fixed some typos in the first version. Furthermore, by your suggestions, we introduce the "effective learning rate" (see Section 4.3 in the revised version).  In the revised version, we show that SL has a worse convergence guarantee than FL in Table 1.
>
> + FL: $\mathcal{O}\left( \frac{F}{\tilde{\eta}\_{\text{FL}}R} \right) + \mathcal{O}\left( \tilde{\eta}^2\_{\text{FL}} G^2 + \frac{\tilde{\eta}^2\_{\text{FL}}\sigma^2}{K} \right) + \mathcal{O}\left( \frac{\tilde{\eta}\_{\text{FL}}\sigma^2}{NK} \right)$,
>   where $\tilde{\eta}\_{\text{FL}} = K\eta_g\eta_l$, $K$ is the number of local update steps, $N$ is the number of clients, $F= f(\mathbf{x}^0)- f(\mathbf{x}^*)$, $\sigma$ is the variance of the stochastic gradients, $G$ measures the data heterogeneity.
>
> + SL: $\mathcal{O}\left( \frac{F}{\tilde{\eta}\_{\text{SL}}R} \right) + \mathcal{O}\left( \tilde{\eta}^2\_{\text{SL}} G^2 + \frac{\tilde{\eta}^2\_{\text{SL}}\sigma^2}{K} \right) + \mathcal{O}\left( \frac{\tilde{\eta}\_{\text{SL}}\sigma^2}{K} \right)$
>   where $\tilde{\eta}\_{\text{SL}} = NK\eta_g\eta_l$.
>
> Experimental results also show that SL has a worse performance than FL when $K$ is large. Our work can bridge the gap between FL and SL, provide deep understanding of these two approaches and guide the deployment of these two in real-world applications.
>
> ##### **Q6. Does K depend on i.**
>
> > Does K depend on i? you have K=En_i/b.
>
> In fact, $K$ can be set to any value without any dependencies. Only in our experiments we use the notation $E$ to measure the value of $K$, since $E$ satisfies $K = \max \\{ En_i/b\\}$ in the experiments, where $b$ is mini-batch size, $n_i$ is the size of local dataset (We have highlighted in Section 5.2 in blue in the revised version). The notation of $K=En_i/b$ in the first version is less rigorous.
>
> ---
>
> We hope that our response clarifies the training process of SL and addresses your concerns. We are glad to answer any further questions. Thank you!
>
> ### References
>
> [1] Thapa, C., Arachchige, P. C. M., Camtepe, S., & Sun, L. (2022, June). Splitfed: When federated learning meets split learning. In *Proceedings of the AAAI Conference on Artificial Intelligence* (Vol. 36, No. 8, pp. 8485-8493).

---

> > ### Comment · Reviewer_SLb6 · 2022-11-25
> > **Comments on the rebuttal**
> >
> > Were there errors in your previous stated complexities in table 2? Otherwise since you agree with my remark, you get the same bounds. In your table 2 you take eta_SL = eta_FL/N you get exactly the same bounds.

---

> > > ### Author Response · Authors · 2022-11-26
> > > **Further response to Reviewer SLb6**
> > >
> > > ##### **Q2: Were there errors in your previous stated complexities in table 2?**
> > >
> > > Yes, but the typo does not affect our convergence rate and we have fixed it in the revised version. In the revised version, we show that SL has a worse convergence guarantee than FL in Table 1.
> > > + FL: $\mathcal{O}\left( \frac{F}{\tilde{\eta}\_{\text{FL}}R} \right) + \mathcal{O}\left( \tilde{\eta}^2\_{\text{FL}} G^2 + \frac{\tilde{\eta}^2\_{\text{FL}}\sigma^2}{K} \right) + \mathcal{O}\left( \frac{\tilde{\eta}\_{\text{FL}}\sigma^2}{NK} \right)$, where $\tilde{\eta}\_{\text{FL}} = K\eta_g\eta_l$.
> > > + SL: $\mathcal{O}\left( \frac{F}{\tilde{\eta}\_{\text{SL}}R} \right) + \mathcal{O}\left( \tilde{\eta}^2\_{\text{SL}} G^2 + \frac{\tilde{\eta}^2\_{\text{SL}}\sigma^2}{K} \right) + \mathcal{O}\left( \frac{\tilde{\eta}\_{\text{SL}}\sigma^2}{K} \right)$, where $\tilde{\eta}\_{\text{SL}} = NK\eta_g\eta_l$.
> > >
> > > It is reasonable for FL and SL have the similar results, since the main issue of FL and SL are both the client-drift [1] problem.
> > >
> > > We hope that our response addresses your concerns. Thank you!
> > >
> > > [1] Karimireddy, S. P., Kale, S., Mohri, M., Reddi, S., Stich, S., & Suresh, A. T. (2020, November). Scaffold: Stochastic controlled averaging for federated learning. In *International Conference on Machine Learning* (pp. 5132-5143). PMLR.

---

> > > > ### Comment · Reviewer_SLb6 · 2022-11-30
> > > > **Comments on the rebuttal**
> > > >
> > > > I saw these new bounds in your previous comments and the updated version of the paper. My concern is:
> > > >  In your table 2 in the first version of the paper if you take eta_SL = eta_FL/N you get exactly the same bounds.
> > > > So since the typos you had in the previous version do not affect the final results, then you should get with these lrs the same bounds for FL and SL.

---

> > > > > ### Author Response · Authors · 2022-11-30
> > > > > **Further response to Reviewer SLb6**
> > > > >
> > > > > The convergence guarantees of SL in the previous version and revised version are summarized as follows:
> > > > > $$
> > > > > \text{previous:\\;}\mathcal{O}\left( \frac{F}{\tilde{\eta}\_{\text{SL}}R} \right) + \mathcal{O}\left( \tilde{\eta}^2\_{\text{SL}} G^2 + \frac{\tilde{\eta}^2\_{\text{SL}}\sigma^2}{K} \right) + \mathcal{O}\left( \frac{\tilde{\eta}\_{\text{SL}}\sigma^2}{NK} \right)
> > > > > $$
> > > > > $$
> > > > > \text{revised:\\;}\mathcal{O}\left( \frac{F}{\tilde{\eta}\_{\text{SL}}R} \right) + \mathcal{O}\left( \tilde{\eta}^2\_{\text{SL}} G^2 + \frac{\tilde{\eta}^2\_{\text{SL}}\sigma^2}{K} \right) + \mathcal{O}\left( \frac{\tilde{\eta}\_{\text{SL}}\sigma^2}{K} \right)
> > > > > $$
> > > > >
> > > > > + For SL, the only difference in these two versions is the third term $\mathcal{O}\left( \frac{\tilde{\eta}\_{\text{SL}}\sigma^2}{NK} \right)$ (previous version) and $\mathcal{O}\left( \frac{\tilde{\eta}\_{\text{SL}}\sigma^2}{K} \right)$ (revised version). So for both versions, by choosing $\eta_g\eta_l = \frac{1}{L\sqrt{T}}$, we can get the convergence rate of $\mathcal{O}\left(\frac{1}{\sqrt{T}}\right)$ (Some detailed information can be omitted when $T$ is sufficiently large, like $N$ and $K$). We refer to $\mathcal{O}\left(\frac{1}{\sqrt{T}}\right)$ as the convergence rate in the previous responses.
> > > > >
> > > > > + However, as you pointed out, it actually affects the final convergence guarantee of SL as shown in the two equations above (the third term). Based on this, we have also updated the conclusion of the comparison (between FL and SL) in the revised version.
> > > > >
> > > > > We hope that our response addresses your concerns. Thanks very much for your further comments for us to improve the paper.

---

> ### Author Response · Authors · 2022-11-22
> **Response to Reviewer SLb6 (Part 1/2)**
>
> Thanks so much for checking our submission carefully. We have revised our paper based on your insightful suggestions and highlighted the major changes in blue. In the following, we response to the weaknesses mentioned and suggestions for clarity below.
>
> ### **The clarification of training process of SL**
>
> As suggested, we have clarified the training process of SL in the revised version. Algo. 1 is updated with more details. We polish the description of Algo. 1. We have also provided the detailed illustration of the SL process in Appendix A (e.g., Figure 4). We believe that the updated version has clarified the scenario of interest of this paper and the detailed process of SL. In detail, we have (see the 1-st to 4-th points):
>
> ##### **Q1. Confusing term "Communication rounds".**
>
> > In Algo 1. I have a problem understanding exactly what is the communication round here, because in step 8 and 10 there is communication between clients and server. You don't count that as a communication round?
>
> > To have the local update in step 12 (local model update) according to algo 1, communication is needed between the server and the client, so what is local here since there is a communication between server and client?...
>
> > It seems that there are "local" communication rounds which are denoted by k=0.1,...K and there are "global" communication rounds which are r=0.1...R can you clarify what this is, because for me the communication happens for both iterations k and r? am i missing something?
>
> Thank you for pointing this out. In fact, we refer to the outer loop ($r=0,\ldots,R-1$) of Algo. 1 as the "communication rounds" in the first version, i.e., "global" communication rounds you mentioned. However, as you pointed, there is communication between clients and the server in the local update (denoted by $k$. We still use *local update* here, please see the 2-nd point), which seems very confusing. So in the revised version, we replace the "communication rounds" with "training rounds" or just "rounds", which means that all the clients complete their local updates.
>
> #### **Q2. "Local update" in SL**
>
> > In step 12 you put, client-side and server-side updates, again I'm confused here, how the server can access the local gradients to do this local update.
>
> Actually, there are two synchronous processes carried out on the client-side and server-side in step 12 of the first version, denoted by
> $$
> \begin{cases}
> 		\text{Client $i$:} &\mathbf{x}\_{c,i}^{(r,k+1)} \leftarrow \mathbf{x}\_{c,i}^{(r,k)} - \eta_l \mathbf{g}\_i(\mathbf{x}\_{c,i}^{(r,k)})\\\\
> 		\text{Server:}&\mathbf{x}\_{s,i}^{(r,k+1)} \leftarrow \mathbf{x}\_{s,i}^{(r,k)} - \eta_l \mathbf{g}\_i(\mathbf{x}\_{s,i}^{(r,k)})
> \end{cases}
> $$
> where $\mathbf{x}_c$ and $\mathbf{x}_s$ denote the client-side model and server-side model respectively --- We have updated in our revised version. In other words, the server has calculated the gradients on the server-side model with the activations received from the client (step 8 in the revised version), so the server need not access the gradients of the client.
>
> At the same time, we should note that the two equations can be written as one equation (step 12 in the first version and Eq. (2), (3) in the revised version) for simplicity (also in our proof), since the client-side model and the server-side model are updated synchronously. We can see the concatenation of the client-side model $\mathbf{x}_c$ and the server-side model $\mathbf{x}_s$ as one complete model denoted by $\mathbf{x}$.
>
> Besides, here we adopt the usual concept of "local update" (i.e., $k=0,\ldots,K-1$ in Algo. 1) in SL [1]. However, it is much different from FL, since *Local update* in SL does not mean the update is made inside the client --- In fact, the local update carried out in the client-side model and server-side model synchronously. In SL [1], it adopts a generalized meaning that making updates over the local dataset.
>
> ##### **Q3. why sample a subset of clients only?**
>
> > In step 3, since in SL things work in parallel, why do you need to sample a subset of clients only?
>
> As suggested, we have clarified the training process of SL. As stated in Algo. 1, the clients update the model in sequence (maybe a typo of "in parallel" in the question). We have two reasons for sampling.
>
> + For cross-device settings. The client selection is critical in the cross-device setting where the number of clients/devices can be up to $10^{10}$. In this case, the sequential update of the clients may result in prohibitive training time, and one must sample a subset of clients for training efficiency.
> + For comparison of FL. The client selection is widely adopted in FL in cross-device settings. For fair comparison, the client selection is adopted in SL each round.

---

> > ### Comment · Reviewer_SLb6 · 2022-11-25
> > **Comments on the rebuttal**
> >
> > Still unclear to me. If my understanding of SL is correct then you have a cutting layer where you split the network, the first part is on the clients and the rest on the server, so basically the loss function can be written as l(x) = f (g(x)), where f is on the clients and g on the server. \nabla l(x) = \nabla g(x) \nabla f(g(x)), so the server doesn't have access to gradients during local training because it doesn't have access to f.

---

> > > ### Author Response · Authors · 2022-11-26
> > > **Further response to Reviewer SLb6**
> > >
> > > ##### **Q1. the server doesn't have access to gradients because it doesn't have access to f.**
> > >
> > > > basically the loss function can be written as l(x) = f (g(x)), where f is on the clients and g on the server. \nabla l(x) = \nabla g(x) \nabla f(g(x)), so the server doesn't have access to gradients during local training because it doesn't have access to f.
> > >
> > > Thanks very much for your detailed description for us to address the concerns. It is nice to follow your understanding that the loss function can be written as $l(x) = f(g(x))$, where $g$ is on the clients and $f$ on the server (see the simple Fig. a below). Next we use this example to show the process of SL.
> > >
> > > $x := [x_c;x_s]$ (adopted in SL research [1,2,3]) denotes the model parameters of the complete model; $x_c$ denotes the model parameters of the client-side model; $x_s$ denotes the model parameters of the server-side model.
> > >
> > > + The computation of the client-side model can be represented as taking an input $\xi$ and producing an activation $z = g(\xi;x_c)$ using the client-side model parameters $x_c$.
> > >
> > > + The computation of the server-side model can be represented as taking a input $z$ and producing the loss $l = f(z;x_s)$ using the server-side model parameters $x_s$.
> > > + The back-propagation of the client-side mode and the server-side model are:
> > >
> > >     $$
> > >     \text{Server:} \quad \frac{\partial l}{\partial x_s} = \frac{\partial f(x_s;z)}{\partial x_s};\frac{\partial l}{\partial z} = \frac{\partial f(z;x_s)}{\partial z}
> > >     $$
> > >     $$
> > >     \text{Client:} \quad \frac{\partial l}{\partial x_c} = \frac{\partial g(x_c;\xi)}{\partial x_c} \cdot \frac{\partial l}{\partial z}
> > >     $$
> > >
> > > + The client-side model and the server-side model make updates with $\frac{\partial l}{\partial x_c}$ and $\frac{\partial l}{\partial x_s}$ respectively.
> > >
> > > So the server can make updates with $\frac{\partial l}{\partial x_s}$ without accessing to $g$.
> > >
> > > Thanks very much for the constructive comments. We will add the clarification in Appendix A in the later submission.
> > >
> > > ___
> > >
> > > $g(x_c)$                 $\qquad\qquad\qquad\qquad$                  $f(x_s)$
> > >
> > > client    $\quad$       $\frac{\partial l}{\partial z}$ <---$\quad$---> $z$     $\quad$        server
> > >
> > > Fig. a
> > >
> > > ---
> > >
> > > [1] Thapa, C., Arachchige, P. C. M., Camtepe, S., & Sun, L. (2022, June). Splitfed: When federated learning meets split learning. In *Proceedings of the AAAI Conference on Artificial Intelligence* (Vol. 36, No. 8, pp. 8485-8493).
> > >
> > > [2] Thapa, C., Chamikara, M. A. P., & Camtepe, S. A. (2021). Advancements of federated learning towards privacy preservation: from federated learning to split learning. In *Federated Learning Systems* (pp. 79-109). Springer, Cham.
> > >
> > > [3] Wang, J., Qi, H., Rawat, A. S., Reddi, S., Waghmare, S., Yu, F. X., & Joshi, G. (2022). FedLite: A Scalable Approach for Federated Learning on Resource-constrained Clients. arXiv preprint arXiv:2201.11865.

---

> > > > ### Comment · Reviewer_SLb6 · 2022-11-30
> > > > **Thanks for the clarifications**
> > > >
> > > > Thanks for these clarifications. I get the idea better now.

---

### Decision · Program_Chairs · 2023-01-20

**Decision:**

Reject

**Justification For Why Not Higher Score:**

This could go either way, depending if the two negative reviewers acknowledge the author response. The paper does have some value, placing it narrowly around the bar for ICLR

**Justification For Why Not Lower Score:**

N/A

**Metareview: Summary, Strengths And Weaknesses:**

The paper studies convergence guarantees of split learning (SL) for non-convex objectives on the realistic case of heterogeneous data. This question is of great interest to the community. The paper compares standard federated (FL) with SL and observed SL can beat FL when data homogeneity is high, but will suffer for more heterogeneous data.

Unfortunately after discussions the impression remained that the paper in its current form remains narrowly below the bar. It could have gone either way. The main remaining concerns are due to the level of novelty (wrt existing results on stepsizes as special cases) and improvements to the presentation of the paper. We are also concerned that the bounds are sometimes loose and greatly affected by stepsizes (i.e., $\eta_l, \eta_g$, which is pointed out by reviewer SLb6, and the authors also admit that the theoretical study of $\eta_g$ is still insufficient.

We hope the detailed feedback helps to strengthen the paper for a future occasion, and are positive for that as the paper definitely has merit.